# UNCOVERING CHALLENGES OF SOLVING THE CONTINUOUS GROMOV-WASSERSTEIN PROBLEM

## ABSTRACT

Recently, the Gromov-Wasserstein Optimal Transport (GWOT) problem has attracted the special attention of the ML community. In this problem, given two distributions supported on two (possibly different) spaces, one has to find the most isometric map between them. In the discrete variant of GWOT, the task is to learn an assignment between given discrete sets of points. In the more advanced continuous formulation, one aims at recovering a parametric mapping between unknown continuous distributions based on i.i.d. samples derived from them. The clear geometrical intuition behind the GWOT makes it a natural choice for several practical use cases, giving rise to a number of proposed solvers. Some of them claim to solve the continuous version of the problem. At the same time, GWOT is notoriously hard, both theoretically and numerically. Moreover, all existing continuous GWOT solvers still heavily rely on discrete techniques. Natural questions arise: to what extent do existing methods unravel the GWOT problem, what difficulties do they encounter, and under which conditions they are successful? Our benchmark paper is an attempt to answer these questions. We specifically focus on the continuous GWOT as the most interesting and debatable setup. We crash-test existing continuous GWOT approaches on different scenarios, carefully record and analyze the obtained results, and identify issues. Our findings experimentally testify that the scientific community is still missing a reliable continuous GWOT solver, which necessitates further research efforts. As the first step in this direction, we propose a new continuous GWOT method which does not rely on discrete techniques and partially solves some of the problems of the competitors.

## 1 INTRODUCTION

Optimal Transport (OT) is a powerful framework that is widely used in machine learning (Montesuma et al., 2023). A popular application of OT is the domain adaptation of various modalities, including images (Courty et al., 2016; Luo et al., 2018; Redko et al., 2019), music transcription (Flamary et al., 2016), color transfer (Frigo et al., 2015), alignment of embedding spaces (Chen et al., 2020; Aboagye et al., 2022). Other applications include generative models (Salimans et al., 2018; Arjovsky et al., 2017), unpaired image-to-image transfer (Korotin et al., 2023b; Rout et al., 2022), etc.

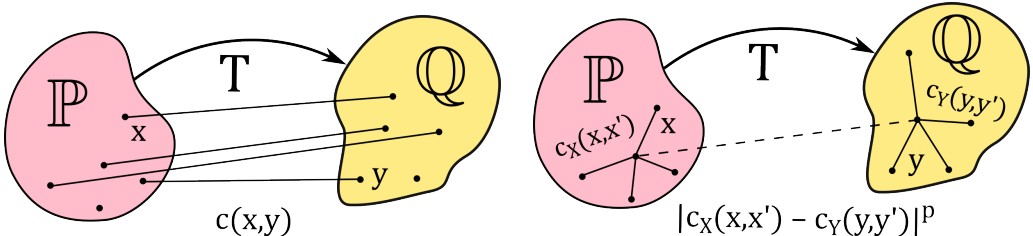

(a) Monge's OT between distributions $\mathbb{P}$ and $\mathbb{Q}$ with *inter-domain* cost function $c(x, y)$.

(b) Monge's GW between distributions $\mathbb{P}$ and $\mathbb{Q}$ with *intra-domain* costs $c_{\mathcal{X}}(x, x')$ and $c_{\mathcal{Y}}(y, y')$.

Figure 1: A schematic visualization of the OT problems and GW problems (Monge's form).

In the conventional OT problem (Figure 1a), one needs to find a map between two data distributions that minimizes a certain "effort" expressed in the form of an *inter-domain* transport cost function.

This cost function shows how hard it is to move a point in the source space to a given point in the target space. Thus, in order for the resulting map to possess certain useful properties, one has to incorporate them into the cost function. Unfortunately, this is not always a straightforward task, especially when the data distributions are supported in different spaces.

A popular way to address the above-mentioned issue is to consider the Gromov-Wasserstein (GW) modification (Mémoli, 2007; 2011; Peyré et al., 2016) of the OT problem (Figure 1b). Here one assumes that both the source and target spaces are equipped with a structure, e.g., with a metric, and one aims to find a transport map that maximally preserves this structure, i.e., the most isometric map. This clear geometrical intuition behind GW makes it natural in various applications: unsupervised data alignment (Alvarez-Melis & Jaakkola, 2018; Aboagye et al., 2022), single-cell data processing (Scetbon et al., 2022; Klein et al., 2023; Sebbouh et al., 2024), 2D and 3D shape analysis (Beier et al., 2022; Mémoli, 2009), graph data analysis (Xu et al., 2019; Vincent-Cuaz et al., 2022; Chowdhury & Needham, 2021; Xu et al., 2021; Vincent-Cuaz et al., 2021).

**Discrete/continuous GW**. The GWOT problem is about learning some specific translation that operates with source and target distributions. In practice, these distributions are typically given by empirical datasets. This leads to two possible ways of paving a GWOT map. In the **discrete** scenario, the learned translation is just a point(s)-to-point(s) assignment (transport matrix). In turn, the **continuous** GW is about learning a parametric mapping between the underlining (continuous) distributions. In this case, the datasets are treated as i.i.d. samples derived from them.

While existing computational approaches for the GW problem show considerable empirical success, the problem itself is highly non-trivial from different perspectives.

- **Theory.** Finding the most isometric map between probability spaces based just on the inner properties of these spaces may be poorly defined, e.g., the desired transform may be non-unique. This happens where the source (target) space permits some non-trivial isometries that preserve the corresponding source (target) distribution. A simple yet expressive example is the Gaussian case Delon et al. (2022). Intuitively, non-uniqueness may affect the stability of a GWOT solver.

- **Computations/algorithms.** It is known that the discrete GW yields a non-convex quadratic optimization problem (Vayer, 2020), which is computationally challenging. To partially alleviate the difficulty, one typical approach is to consider entropic regularization (Peyré et al., 2016; Alvarez-Melis & Jaakkola, 2018; Scetbon et al., 2022; Wang & Goldfeld, 2023). Fortunately, the regularized problem resorts to a sequence of tractable Sinkhorn OT assignments. However, the convergence of the procedure may not hold, see (Peyré et al., 2016, Remark 3). In addition, discrete GWOT scales poorly with the number of input (source or target) samples, which makes some problem setups unmanageable by such kinds of solvers. While there are some techniques to reduce the computational burden w.r.t data size (Scetbon et al., 2022), they come at the cost of additional restrictions and assumptions.

- **Methodology.** The majority of existing continuous GW methods are based on discrete GW techniques and inherit all the computational challenges of the latter. Moreover, the transition from the discrete to the continuous setup may be questionable from a statistical point of view (Zhang et al., 2024).

Having said that, one naturally wonders: how do the current continuous GWOT methods manage to overcome these problems and show good practical results? What are the "bad cases" under which the aforementioned difficulties become critical and the solvers struggle? How to fight with these "bad cases"? In our paper, we shed light on these GWOT methods' ambiguities, specifically focusing on the continuous setup. **Our contributions** are as follows:

- We conduct a deep analysis of existing papers and reveal that one important characteristic that may greatly affect practical performance is the considered data setup. In fact, the majority of works primarily consider datasets with some specific correlations between source and target samples. Formally speaking, such setups disobey the standard i.i.d. assumption on the data and may lead to spoiled conclusions on the solvers' capabilities.

- By following the findings from the previous point, we evaluate the performance of existing continuous GWOT solvers in more statistically fair and practically realistic *uncorrelated* data setups. Our simple yet expressive experiments witness that *(un)correlatedness* indeed highly influences the quality of the learned GWOT maps. Changing the data setup may greatly deteriorate the performance of the solvers.

- To alleviate the dependence on the mutual statistical characteristics of the source and target training data, we propose a novel continuous neural GW solver. On the one hand, our method is not based on

discrete GW. It may be learned on arbitrarily large datasets and shows reasonably good results even on the fair *uncorrelated* data setup. On the other hand, the method is min-max-min adversarial, which negatively impacts stability and requires plenty of data for training.

Overall, our findings reveal that the empirical success of the existing GWOT solvers seems to be a bit over-estimated and requires to be treated more critically. Constructing a reliable continuous GWOT method is a not-yet-solved challenge. We encourage the researchers to further work out on this quite interesting direction. We hope, that our work is a good amigo in this thorny path.

**Notations.** Throughout the paper, $\mathbb{R}^{d_x}$ and $\mathbb{R}^{d_y}$ are the source and target data spaces, respectively. The set of Borel probability distributions on $\mathbb{R}^{d_x}$ is $\mathcal{P}(\mathbb{R}^{d_x})$. The dot product of vectors $\mathbf{x}, \mathbf{x}' \in \mathbb{R}^{d_x}$ is $\langle \mathbf{x}, \mathbf{x}' \rangle_{d_x}$. For a measurable map $T \colon \mathbb{R}^{d_x} \to \mathbb{R}^{d_y}$, we denote the corresponding *push-forward* operator by $T_\sharp$. For $\mathbb{P} \in \mathcal{P}(\mathbb{R}^{d_x})$ and $\mathbb{Q} \in \mathcal{P}(\mathbb{R}^{d_y})$, we denote the set of all couplings between them by $\Pi(\mathbb{P}, \mathbb{Q})$, i.e., distributions $\pi$ on $\mathbb{R}^{d_x} \times \mathbb{R}^{d_y}$ with the corresponding marginals equal to $\mathbb{P}$ and $\mathbb{Q}$.

## 2 BACKGROUND

In this section, we first explain the conventional OT setup (Villani, 2008; Santambrogio, 2015; Gozlan et al., 2017; Backhoff-Veraguas et al., 2019) and then introduce the Gromov-Wasserstein OT formulation (Mémoli, 2011; Peyré et al., 2016). Finally, we clarify our considered practical learning setup under which these problems are considered.

### 2.1 OPTIMAL TRANSPORT (OT) PROBLEM

Given two probability distributions $\mathbb{P} \in \mathcal{P}(\mathbb{R}^{d_x})$, $\mathbb{Q} \in \mathcal{P}(\mathbb{R}^{d_y})$ and a cost function $c \colon \mathbb{R}^{d_x} \times \mathbb{R}^{d_y} \to \mathbb{R}$, the OT problem is defined as follows:

$$\mathrm{OT}_c(\mathbb{P}, \mathbb{Q}) \stackrel{\text{def}}{=} \inf_{T_\sharp \mathbb{P} = \mathbb{Q}} \int_{\mathbb{R}^{d_x}} c(\mathbf{x}, T(\mathbf{x})) d\mathbb{P}(\mathbf{x}). \tag{1}$$

This is known as Monge's formulation of the OT problem. Intuitively, it can be understood as finding an optimal transport map $T^* \colon \mathbb{R}^{d_x} \to \mathbb{R}^{d_y}$ that transforms $\mathbb{P}$ to $\mathbb{Q}$ and minimizes the total transportation expenses w.r.t. cost $c$, see Figure 1a. There have been developed a lot of methods for solving OT (1) in its discrete (Cuturi, 2013; Peyré et al., 2019) and continuous (Makkuva et al., 2020; Daniels et al., 2021; Korotin et al., 2023b; Choi et al., 2023; Fan et al., 2023; Uscidda & Cuturi, 2023; Gushchin et al., 2024; Mokrov et al., 2024; Asadulaev et al., 2024) variants.

The cost function $c$ in (1) is commonly the squared Euclidean distance. In this case, problem (1) is exclusively defined for spaces of the same dimensions. Dealing with two incomparable spaces ($d_x \neq d_y$) may require manually defining some more complex inter-domain cost function $c$. It is not a trivial task.

### 2.2 GROMOV-WASSERSTEIN OT (GWOT) PROBLEM

The GWOT problem is an extension of the optimal transport problem that aims to compare and transport probability distributions supported on different spaces. This problem is particularly useful when the underlying spaces do not align directly, but we still want to measure and align their intrinsic geometric structures. In what follows, we introduce the discrete and continuous variants of GWOT.

**Discrete Gromov-Wasserstein formulation.** Let $N_x$ and $N_y$ be the number of training samples in the source and target domains, respectively. Let $\mathbf{C}^x \in \mathbb{R}^{N_x \times N_x}$ and $\mathbf{C}^y \in \mathbb{R}^{N_y \times N_y}$ be the corresponding source and target intra-domain cost matrices. These matrices measure the pairwise distance or similarity between the samples for a given function, i.e., cosine similarity, Euclidean distance, inner product, etc. The discrete GWOT problem is defined as:

$$\mathbf{T}^* \stackrel{\text{def}}{=} \underset{\mathbf{T} \in \mathcal{C}_{N_x, N_y}}{\arg\min} \sum_{i,j,k,l} |\mathbf{C}_{i,k}^x - \mathbf{C}_{j,l}^y|^p \mathbf{T}_{i,j} \mathbf{T}_{k,l}, \tag{2}$$

where $\mathcal{C}_{N_x, N_y} \stackrel{\text{def}}{=} \{\mathbf{T} \in \mathbb{R}_+^{N_x \times N_y} \mid \mathbf{T}^T \mathbb{1}_{N_x} = \frac{1}{N_y} \mathbb{1}_{N_y}; \mathbf{T} \mathbb{1}_{N_y} = \frac{1}{N_x} \mathbb{1}_{N_x}\}$ is the set of coupling matrices between source and target spaces; $\mathbb{1}_N = [1, \ldots, 1]^T \in \mathbb{R}^N$. The loss function $|\cdot - \cdot|^p$ in (2) is used to account for the misfit between the similarity matrices, a typical choice for the degree factor is $p = 2$ (quadratic loss). Further details can be found in (Peyré et al., 2016; Mémoli, 2011; Chowdhury & Mémoli, 2019; Titouan et al., 2019b).

**Continuous Gromov-Wasserstein formulation.** Let $\mathbb{P} \in \mathcal{P}(\mathbb{R}^{d_x})$, $\mathbb{Q} \in \mathcal{P}(\mathbb{R}^{d_y})$ be two distributions. Let $c_\mathcal{X} : \mathbb{R}^{d_x} \times \mathbb{R}^{d_x} \to \mathbb{R}$ and $c_\mathcal{Y} : \mathbb{R}^{d_y} \times \mathbb{R}^{d_y} \to \mathbb{R}$ be two intra-domain cost functions for the source ($\mathbb{R}^{d_x}$) and target ($\mathbb{R}^{d_y}$) domains, respectively. The Monge's GWOT problem is defined as:

$$\mathrm{GWOT}_p^p(\mathbb{P}, \mathbb{Q}) \overset{\text{def}}{=} \inf_{T_\sharp \mathbb{P} = \mathbb{Q}} \int_{\mathbb{R}^{d_x}} \int_{\mathbb{R}^{d_x}} |c_\mathcal{X}(\mathbf{x}, \mathbf{x}') - c_\mathcal{Y}(T(\mathbf{x}), T(\mathbf{x}'))|^p \, d\mathbb{P}(\mathbf{x}) d\mathbb{P}(\mathbf{x}'). \tag{3}$$

Theoretical results on the existence and regularity of (3) under certain cases could be found in (Dumont et al., 2024; Mémoli & Needham, 2024; 2022). An intuitive illustration of problem (3) can be found in Figure 1b. In this continuous setup, the objective is to find an optimal transport map $T^* : \mathbb{R}^{d_x} \to \mathbb{R}^{d_y}$ that allows to transform (align) the source distribution to the target distribution. While in (1) we search for a map that sends $\mathbb{P}$ to $\mathbb{Q}$ minimizing the total transport cost, (3) aims to find the most isometric map w.r.t. the costs $c_\mathcal{X}$ and $c_\mathcal{Y}$, i.e., the map that maximally preserves the pairwise intra-domain costs. The commonly studied case (Vayer, 2020; Sebbouh et al., 2024) is $p = 2$ with the Euclidean distance $c(\cdot, \cdot) = \| \cdot - \cdot \|^2$ or inner product $c(\cdot, \cdot) = \langle \cdot, \cdot \rangle$ as intra-domain cost functions. In what follows, we will use **innerGW** to denote the latter case.

### 2.3 PRACTICAL LEARNING SETUP

In practical scenarios, the source and target distributions $\mathbb{P}$ and $\mathbb{Q}$ are typically accessible by empirical samples (datasets) $X = \{\mathbf{x}_i\}_{i=1}^{N_x} \sim \mathbb{P}$ and $Y = \{\mathbf{y}_i\}_{i=1}^{N_y} \sim \mathbb{Q}$. Under the **discrete** GWOT formulation, these samples are directly used to compute intra-domain cost matrices $\mathbf{C}^x$, $\mathbf{C}^y$. These matrices are then fed to optimization problem (2). Having been solved, problem (2) yields a coupling matrix $\mathbf{T}^*$ which defines the GWOT correspondence between $X$ and $Y$. Importantly, discrete GWOT operates with discrete empirical measures $\hat{\mathbb{P}} \overset{\text{def}}{=} \sum_{i=1}^{N_x} \frac{1}{N_x} \delta(\mathbf{x} - \mathbf{x}_i)$, $\hat{\mathbb{Q}} \overset{\text{def}}{=} \sum_{i=1}^{N_y} \frac{1}{N_y} \delta(\mathbf{y} - \mathbf{y}_i)$ rather than original ones. In turn, under the **continuous** formulation, the aim is to recover some parametric map $T^* : \mathbb{R}^{d_x} \to \mathbb{R}^{d_y}$ between the original source and target distributions $\mathbb{P}$ and $\mathbb{Q}$. In most practical scenarios, the latter is preferable, as it naturally allows *out-of-sample* estimation, i.e., provides GWOT mapping for new (unseen) samples $\mathbf{x} \sim \mathbb{P}$. In our paper, we deal with continuous setup.

## 3 EXISTING CONTINUOUS GROMOV-WASSERSTEIN SOLVERS

Here we outline the current progress in solving the GWOT problem specifically focusing on the continuous formulation. Most of the GWOT solvers are only discrete or adapted to *emulate a continuous behaviour* by implementing some specific out-of-sample estimation method on top of the results of some discrete solver. The initial approach to solve the GWOT problem in discrete case (§2.2) was introduced in (Mémoli, 2011; Peyré et al., 2016). Below we only detail the methods which specifically aim to solve the continuous formulation and somehow provide the out-of-sample estimation.

**StructuredGW (Sebbouh et al., 2024)**. In this paper, the authors focus on providing an iterative algorithm to solve a discrete entropy-regularized version of the inner product case of (3) for $p = 2$ using the equivalent reformulation by (Vayer, 2020, maxOT). Every iteration, the coupling matrix $\mathbf{T}$ is updated using Sinkhorn iterations and an auxiliary rotation matrix updates using different possible methods. The authors propose different regularization alternatives for the problem, this directly impacts the way the auxiliary matrix is updated. To perform the out-of-sample estimation, the authors extend their method by drawing inspiration from entropic maps in (Pooladian & Niles-Weed, 2024; Dumont et al., 2024). Their developed StructuredGW method uses one of the dual potentials learned during the updates of $\mathbf{T}$ to perform this entropic mapping.

**FlowGW (Klein et al., 2023).** The framework proposed in this work consists in fitting a discrete GW solver inspired by (Peyré et al., 2016) to obtain a coupling matrix $\mathbf{T}$. This coupling matrix helps to figure out the best way to match available samples from source to target domains. Weighted pairs of source and target samples are constructed using the distribution described by the coupling matrix. Then these samples are used to train a Conditional Flow Matching (CFM) model (Lipman et al., 2023) with the noise outsourcing technique from (Kallenberg, 1997). The inference process is done by solving the ODE given by the CFM model.

**AlignGW (Alvarez-Melis & Jaakkola, 2018)**. This work proposes a discrete solver for the alignment of word embeddings. The authors use Sinkhorn iterations to compute the updates on the coupling matrix instead of a linear search implemented in the Python Optimal Transport library (Flamary et al., 2021) that is inspired by (Titouan et al., 2019a; Peyré et al., 2016). This change significantly improves the stability of the solver. In spite of being a totally discrete method, we consider it due to its

performance in challenging tasks. In order to allow out-of-sample estimations, we train a Multi-Layer Perceptron (MLP) model on the barycentric projections derived from the learned coupling matrix **T**.

**RegGW (Uscidda et al., 2024)**. This work extends the Monge Gap Regularizer (Uscidda & Cuturi, 2023) within the Gromov-Wasserstein (Monge) framework. The method parameterizes the transport push-forward function $T$ using a neural network. To train this model, a regularized loss function is utilized, which combines *fitting* loss and Gromov-Monge gap *regularizer*. The *fitting* loss ensures that the learned model $T$ maps to the target distribution. It is chosen to be Sinkhorn divergence in practice (Uscidda et al., 2024). In turn, the Gromov-Monge gap is the difference between *distortion* — the value of discrete Gromov-Wasserstein functional for learned model $T$, and the actual solution of discrete GW problem between source points and points mapped with $T$. While other research, such as (Sotiropoulou & Alvarez-Melis, 2024), also incorporates this regularizer, their approach requires access to an intermediate reference distribution, which is impractical for our experimental setup.

**CycleGW (Zhang et al., 2021)**. The authors of this work propose to minimize the Unbalanced bidirectional Gromov–Monge divergence (UBGMD) and recover two push-forward Gromov-Monge mappings: $f$ such that $f_\sharp \mathbb{P} \approx \mathbb{Q}$ and $g$ such that $g_\sharp \mathbb{Q} \approx \mathbb{P}$. This problem is similar in nature to the Unbalanced Gromov-Wasserstein divergence Séjourné et al. (2020), but it utilizes a cross-domain version of (3) which additionally ensures cycle-consistency. To solve the UBGMD problem they propose to minimize so-called Generalized Maximum Mean Discrepancy (GMMD), in which they compute the divergences of the unbalanced problem by using Maximum Mean Discrepancy (MMD) with Gaussian kernels. In our setup, the function $f$ is equivalent to our mapping function $T$. We refer to the original work for further details. A work guided by a similar concept can be found in Hur et al. (2021) which adds an additional MMD term. Due to the similarities between these two solvers, we decided to only consider the work by Zhang et al. (2021), they have publicly available code.

### 3.1 TOY 3D→2D EXPERIMENT

To illustrate the capabilities of the solvers and as a necessary sanity check for the implementations, we propose a toy experiment. In this setup, the source distribution is a mixture of Gaussians in $\mathbb{R}^3$ and the target is also a mixture of Gaussians but in $\mathbb{R}^2$, see Figure 2a. By choosing this experiment on incomparable spaces, we ensure the solvers are actually capable of dealing with a real Gromov-Wasserstein problem. The obtained results for the baselines can be found in Figure 2d and 2c, Figure 2b shows the result for our method, NeuralGW. As we can see, for all methods a component of the source distribution is mostly mapped to a component/neighbouring components of the target distribution, indicating the correct GW alignment.

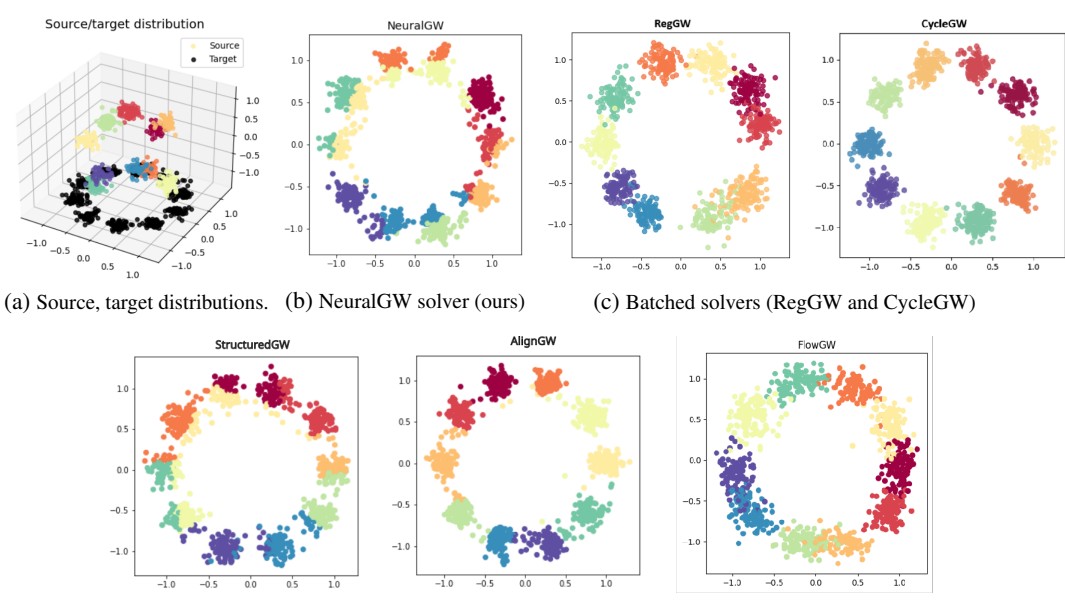

(a) Source, target distributions.  (b) NeuralGW solver (ours)  (c) Batched solvers (RegGW and CycleGW)

(d) Baseline solvers (StructuredGW, AlignGW, FlowGW).

Figure 2: Learned GWOT map $T$ by different solvers; Toy (3D→2D) experiment.

# 4 LIMITATIONS OF EXISTING METHODS

As it was mentioned before, the majority of existing Gromov-Wasserstein approaches explicitly or implicitly resort to discrete GWOT formulation, see §2.2. Naturally, the computations required to solve (2) increase significantly as the numbers of training samples $N_x$ and $N_y$ grow. This dependency renders some datasets to hardly be manageable by discrete solvers. However, in our paper, we specifically focus on the other sources of potential failures for GWOT. It stems from the practical data setups under which the methods actually work. Below, we give a detailed description of this problem.

## 4.1 PITFALLS OF PRACTICAL DATA SETUP

To begin with, we introduce the notion of **(un)correlatedness** of data which undergoes Gromov-Wasserstein alignment. To fit a GWOT solver, a practitioner typically has two training datasets, $X = \{\mathbf{x}_i\}_{i=1}^{N_x}$ and $Y = \{\mathbf{y}_i\}_{i=1}^{N_y}$. They are sampled from the reference source ($\mathbb{P}$) and target ($\mathbb{Q}$) distributions, see our training setup, §2.3. In what follows, without loss of generality, we will assume $N_x = N_y \overset{\text{def}}{=} N$. The natural statistical assumption on samples $X$ and $Y$ is that they are mutually independent. We define this data setup as *uncorrelated*. Simultaneously, we introduce an alternative setup under which the source and target datasets $X$ and $Y$ turn out to be connected by some specific statistical relationships. Let us assume that there is a coupling $\pi \in \Pi(\mathbb{P}, \mathbb{Q})$, $\pi \neq \mathbb{P} \otimes \mathbb{Q}$. Practically, we expect that samples from the coupling are meaningfully dependent, i.e., $\pi$ is "significantly" different from the independent coupling $\mathbb{P} \otimes \mathbb{Q}$. In particular, coupling $\pi$ may even set a one-to-one correspondence between the domains. Then, we call the training samples $X$ and $Y$ to be *correlated* if they are obtained with the following procedure:

1. First, we jointly sample $X$ and $\widetilde{Y} = \{\widetilde{\mathbf{y}}_i\}_{i=1}^N \subset \mathbb{R}^{d_y}$ from coupling $\pi$, i.e.: $X \times \widetilde{Y} \sim \pi$.

2. Secondly, we apply *an unknown permutation $\sigma$* of indices to $\widetilde{Y}$ yielding $Y$, i.e.: $Y = \sigma \circ \widetilde{Y}$.

We found that the majority of the experimental test cases, on which the existing GWOT solvers are validated, frequently follow exactly the *correlated* data setup. For instance, in the problem of learning cross-lingual word embedding correspondence (Alvarez-Melis & Jaakkola, 2018; Grave et al., 2019), the underlining coupling $\pi$ could be understood as the distribution of dictionary pairs. The other example is the bone marrow dataset (Luecken et al., 2021; Klein et al., 2023). In this case, the source and target samples are generated using two different methods to profile gene expressions on the same donors. Interestingly, the *correlated* data setup suits well the discrete GWOT formulation, because the optimization problem in this case boils down to the search for the permutation $\sigma$ that spawned the target dataset $Y$. This leads to the natural **hypothesis** that for *uncorrelated* training datasets $X$ and $Y$ the performance of existing GWOT solvers may be poor. To test this hypothesis, we propose the experimental framework described in the next paragraphs.

**Modeling (un)correlatedness in practice.**
Let $X = \{\mathbf{x}_1, \ldots, \mathbf{x}_N\} \sim \mathbb{P}$ and $Y = \{\mathbf{y}_1, \ldots, \mathbf{y}_N\} \sim \mathbb{Q}$ be the source and target datasets. We assume that $X$ and $Y$ are totally paired, i.e., every $i$-th vector in the source set $X$ is the pair of the $i$-th vector in the target set, $Y$. Also, we suppose that the pairing is reasonable, i.e., dictated by the nature of the data on hand. For example, if $X$ and $Y$ are word embeddings, then $\mathbf{x}_i$ and $\mathbf{y}_i$ correspond to the same word. We propose a way how to create training data with different levels of correlatedness. Initially, batches of $N$ paired (source and target) samples are randomly selected from the datasets, then split into train and test sets, $N = N_{train} + N_{test}$. The train samples are then divided into two halves and a value $\alpha$, $0 \leq \alpha \leq 1$ is set, this value represents the fraction of $N_{train}/2$ samples that will be paired.

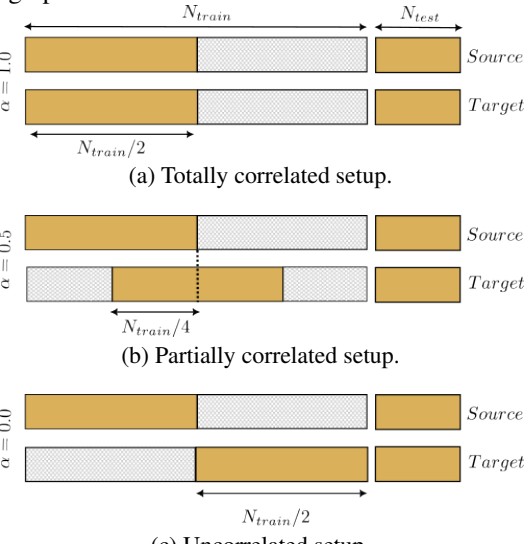

(a) Totally correlated setup.

(b) Partially correlated setup.

(c) Uncorrelated setup.

Figure 3: Data splitting and (un)correlatedness.

The resulting training datasets (both source and target) will totally contain $N_{train}/2$ samples. They are formed by selecting specific indices from the original train sets. Indices from 0 to $N_{train}/2$ are taken from the source while indices from target

are shifted, we take the $\lceil (1-\alpha)(N_{train}/2) \rceil$ to $\lceil (1-\alpha/2)N_{train} \rceil$ indices. As a result, setting a value of $\alpha = 1$ will represent a *totally correlated* setup (Figure 3a), $0 < \alpha < 1$ is *partially correlated* (Figure 3b) and $\alpha = 0$ is *uncorrelated* (Figure 3c).

To conclude the subsection, we want to emphasize that both *correlated* and *uncorrelated* setups are practically important. Some real-world use cases of the former include word embedding assignment and gene expression profiles matching problems, see the details in the text above. In turn, the *uncorrelated* setup naturally appears when aligning single-cell multi-omics data (Demetci et al., 2022). Here one aims at matching different single-cell assays, which are uncorrelated, because applying multiple assays on the same single-cell is typically impossible.

## 4.2 BENCHMARKING GWOT SOLVERS ON (UN)CORRELATED DATA: GLOVE AND BPEMB EXPERIMENTS

In order to check how existing continuous GW solvers perform under uncorrelated, partially and totally correlated setups, we utilize two different text corpora: Twitter and MUSE (Multilingual Unsupervised and Supervised Embeddings) (Conneau et al., 2017) bilingual vocabularies. We then embed them using either the GloVe (Global Vectors for Word Representation) algorithm (Pennington et al., 2014) or BPEmb (Byte-Pair) (Heinzerling & Strube, 2018) embeddings. The Twitter corpus used was obtained from the GloVe dictionary[1]. In the case of MUSE we took the bilingual vocabularies from their official GitHub repository[2]

The GloVe embeddings of words are generated using the GloVe algorithm (Pennington et al., 2014). Its main advantage is that the embedded vectors capture semantic relationships and exhibit linear substructures in the vector space. This allows meaningful computation of distances and alignments, which is central to the GWOT framework. The authors provide access to the GloVe embeddings of four datasets: Wikipedia, Gigaword, Common Crawl, and Twitter. For the experiments in this section, we focus solely on the GloVe embeddings for the Twitter corpus. From this point onward, we will refer to this combination as the "Twitter-GloVe dataset", this notation will also be used in the future to denote other corpus and type of embedding combinations. Alternatively, we explore the use of Byte-Pair embeddings (BPEmb), which is a subword tokenization method that breaks words into smaller units. It works by iteratively merging the most common pairs of adjacent symbols (like characters or character groups) in a corpus until a set vocabulary size is reached. We refer to Appendix B.2 for additional insights and experiments for BPEmb on Twitter and the MUSE corpus.

The paired samples of the Twitter-GloVe dataset are constructed by picking the first 400K word embeddings from a total of around 1.2 million, we refer to it as our "whole" data space. The following blends of dimensionalities are considered: $100 \to 50$, $50 \to 100$, $50 \to 25$ and $25 \to 50$. We test **three** baseline solvers introduced in §3: StructuredGW, AlignGW, FlowGW. We fit every solver for different values of $\alpha$ (different levels of correlatedness). The values of $\alpha$ range from 0.0 to 1.0 in increments of 0.1. For each value of $\alpha$, we perform **ten** fitting repetitions with different random seeds following the process described in §4.1. We use $N_{train} = 6K$, i.e., every experiment run exploits source and target datasets containing $N_{train}/2 = 3K$ training samples; $N_{test} = 2048$. Note that the only reason why we choose such a relatively small size for the training datasets is the computational complexity of the three solvers under consideration. The discrete optimization procedures run at the backend of baseline solvers hardly could be adopted to reasonably larger values of $N_{train}$.

Regarding RegGW and GycleGW baselines from §3, their training fails for the small number of training samples, e.g., 3K. These methods are left to §5.2, where much larger datasets are considered.

For evaluation, we measure Top $k$-accuracy $\uparrow$, cosine similarity $\uparrow$ and FOSCTTM $\downarrow$, the details are given in Appendix B.1. The metrics are computed on (unseen) test data with the *reference* points given by the combination of train and test datasets, $N = N_{train} + N_{test} = 8048$. The results of the experiments for the combinations $100 \to 50$ and $50 \to 25$ are shown in the plots below, Figure 4. For additional experimental results on other dimension pairs, see Appendix B.

**Conclusions.** The results indicate that all the baseline solvers perform well in totally correlated scenarios, even when evaluated on unseen data. This demonstrates their ability to learn and capture the inner structures when the data is highly correlated. However, their performance drops significantly as the value of $\alpha$ decreases. We conjecture that the observed behaviour is mainly due to the small sizes of training sets dictated by the discrete nature of the solvers. Indeed, relatively small samples

---

[1] https://radimrehurek.com/gensim/downloader.html

[2] https://github.com/facebookresearch/MUSE

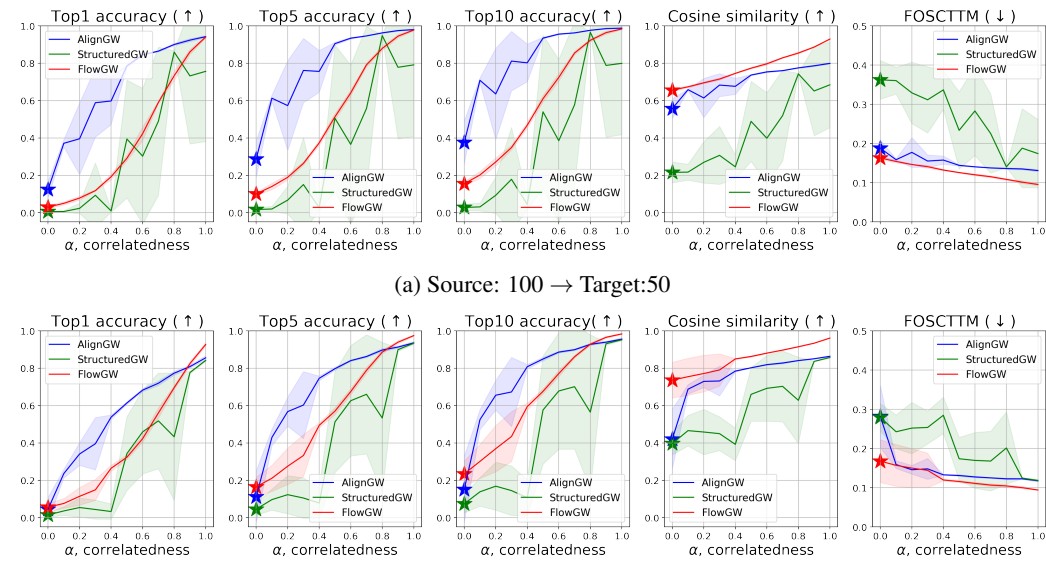

(a) Source: $100 \rightarrow$ Target:50

(b) Source: $50 \rightarrow$ Target:25

Figure 4: Performance of the baseline GWOT solvers for the **Twitter-GloVe** embeddings at different levels of correlatedness $\alpha$ in all **high-to-low** setups. The solvers were trained with $N_{train}/2 = 3000$ samples from a whole space of 400K, this plot shows results for a testing subset of 2048 samples, the metrics were computed considering the $N_{train} + N_{test} = 8048$ samples *reference* space.

hardly could fully express the intrinsic geometry of the data, which complicates the faithful GW alignment of the domains. The broader discussion on the issue of discrete methods under low correlatedness scenario is in Appendix D.1. Therefore, one possible way to increase the performance is to consider GW solvers adapted to large amount of data. In subsequent section (§5), we check different possibilities. In particular, we propose a new continuous solver (§5.1) which does not rely on discrete techniques. Therefore, it can better capture the inner geometry and structure of the data without the strict need of training on correlated data as well as allowing training on large datasets.

## 5 GWOT SOLVERS AT LARGE SCALE

In this section, we start by introducing NeuralGW, a novel scalable method to solve the continuous GWOT problem (§5.1). Then we proceed to the practical performance of NeuralGW and the baselines in large-scale GloVe benchmark (§5.2).

### 5.1 NEURAL GROMOV-WASSERSTEIN SOLVER

In this subsection, we conduct the theoretical and algorithmic derivation of our proposed approach. In what follows, we restrict to the case $d_x \geq d_y$; source ($\mathbb{P}$) and target ($\mathbb{Q}$) distributions are absolutely continuous and supported on some compact subsets $\mathcal{X} \subset \mathbb{R}^{d_x}$, $\mathcal{Y} \subset \mathbb{R}^{d_y}$ respectively. Our method is developed for innerGW, i.e., problem (3) with $c_{\mathcal{X}} = \langle \cdot, \cdot \rangle_{d_x}$, $c_{\mathcal{Y}} = \langle \cdot, \cdot \rangle_{d_y}$, $p = 2$:

$$\text{innerGW}_2^2(\mathbb{P}, \mathbb{Q}) \stackrel{\text{def}}{=} \min_{T_\sharp \mathbb{P} = \mathbb{Q}} \int_{\mathbb{R}^{d_x}} \int_{\mathbb{R}^{d_x}} \left| \langle \mathbf{x}, \mathbf{x}' \rangle_{d_x} - \langle T(\mathbf{x}), T(\mathbf{x}') \rangle_{d_y} \right|^2 d\mathbb{P}(\mathbf{x}) d\mathbb{P}(\mathbf{x}'). \quad (4)$$

Note that the existence of minimizer for (4) is due to (Dumont et al., 2024, Theorem 3.2). We base our method on the theoretical insights about GW from (Vayer, 2020). According to (Vayer, 2020, Theorem 4.2.1), when $\int \|\mathbf{x}\|_2^4 d\mathbb{P}(\mathbf{x}) < +\infty$, $\int \|\mathbf{y}\|_2^4 d\mathbb{Q}(\mathbf{y}) < +\infty$, problem (4) is equivalent to

$$\text{innerGW}_2^2(\mathbb{P}, \mathbb{Q}) = \text{Const}(\mathbb{P}, \mathbb{Q}) - \max_{\pi \in \Pi(\mathbb{P}, \mathbb{Q})} \max_{P \in F_{d_x, d_y}} \int \langle \mathbf{P}\mathbf{x}, \mathbf{y} \rangle_{d_y} d\pi(\mathbf{x}, \mathbf{y}), \quad (5)$$

where $F_{d_x, d_y} \stackrel{\text{def}}{=} \{ \mathbf{P} \in \mathbb{R}^{d_x \times d_y} \mid \|\mathbf{P}\|_{\mathcal{F}} = \min(\sqrt{d_x}, \sqrt{d_y}) \}$ are the matrices of fixed Frobenius norm. Note that (5) admits a solution $\pi^* \in \Pi(\mathbb{P}, \mathbb{Q})$, $P^* \in F_{d_x, d_y}$ (Vayer, 2020, Lemmas 6.2.7; 4.2.2).

Our following lemma reformulates the innerGW problem as a minimax optimization problem. This

reformulation is inspired by well-celebrated dual OT solvers such as (Korotin et al., 2021a; Fan et al., 2023; Korotin et al., 2023b).

**Lemma 5.1 (InnerGW as a minimax optimization)** *It holds that (5) is equivalent to*

$$\text{innerGW}_2^2(\mathbb{P}, \mathbb{Q}) = \text{Const}\,(\mathbb{P}, \mathbb{Q}) + \min_{P \in F_{d_x, d_y}} \max_{f : \mathbb{R}^{d_y} \to \mathbb{R}} \min_{T : \mathbb{R}^{d_x} \to \mathbb{R}^{d_y}} \mathcal{L}(\boldsymbol{P}, f, T), \tag{6}$$

*where*

$$\mathcal{L}(\boldsymbol{P}, f, T) = \int_{\mathbb{R}^{d_y}} f(\boldsymbol{y}) d\mathbb{Q}(\boldsymbol{y}) - \int_{\mathbb{R}^{d_x}} \left[ \langle \boldsymbol{Px}, T(\boldsymbol{x}) \rangle_{d_y} + f(T(\boldsymbol{x})) \right] d\mathbb{P}(\boldsymbol{x}).$$

The following theorem provides a theoretical foundation that validates the minimax optimization framework for solving the GW problem. It shows that under certain conditions, the solution $T^*$ of the minimax problem (4) brings an optimal GW mapping.

**Theorem 5.2 (Optimal maps solve the minimax problem)** *Assume that there exists at least one GW map $T^*$. For any matrix $\boldsymbol{P}^*$ and any potential $f^*$ that solve (6), i.e.,*

$$\boldsymbol{P}^* \in \operatorname*{arg\,min}_{P \in F_{d_x, d_y}} \max_{f} \min_{T : \, \mathbb{R}^{d_x} \to \mathbb{R}^{d_y}} \mathcal{L}(P, f, T) \quad and \quad f^* \in \operatorname*{arg\,max}_{f} \min_{T : \, \mathbb{R}^{d_x} \to \mathbb{R}^{d_y}} \mathcal{L}(\boldsymbol{P}^*, f, T),$$

*and for any GW map $T^*$, we have:*

$$T^* \in \operatorname*{arg\,min}_{T : \, \mathbb{R}^{d_x} \to \mathbb{R}^{d_y}} \mathcal{L}(\boldsymbol{P}^*, f^*, T). \tag{7}$$

To optimize 6 we follow the best practices from the field of continuous OT (Korotin et al., 2021b; Fan et al., 2023; Korotin et al., 2023a;b; Asadulaev et al., 2024; Choi et al., 2023; Gushchin et al., 2024) and simply parameterize $f$ and $T$ with neural networks. In turn, **P** is a learnable matrix of fixed Frobenius norm. We use the alternating stochastic gradient ascent/descent/ascent method to train their parameters. The learning algorithm is detailed in the Appendix C. We call the method NeuralGW. As the sanity check, we run our proposed approach on toy setup from §3.1, see Figure 2b. Note that in comparison to other continuous GWOT approaches, our method *does not rely on discrete OT* in any form. In particular, the training process assumes access to just random samples from $\mathbb{P}, \mathbb{Q}$; it does not use/need any pairing between them.

### 5.2 PRACTICAL PERFORMANCE OF NEURALGW AND BASELINES AT LARGE SCALE

We start by introducing our large-scale Twitter-GloVe setup. We consider the same data preparation process as in §4.2, but take $N_{train} = 360K$ samples instead of the 6K used in §4.2 for the baseline solvers; $N_{test} = 2048$ is left the same. Therefore, each repetition consists in training the models with the same data (180K samples) but using different initialization parameters for, e.g., the neural networks. As the *reference* dataset for metrics computation, we used the whole dataset of Twitter-GloVe embeddings (400K samples).

The competitive methods for the comparison under large-scale Twitter-GloVe setups are: NeuralGW (§5.1); RegGW (§3); CycleGW (§3) and FlowGW (§3). The latter is trained in a minibatch manner, i.e., it fits flow matching on top of discrete GW solutions for minibatches. For completeness, we additionally report the performance of the baselines from §4.2 (in gray, labelled as "Other methods", Figure 5). Note that they are trained on a small subset ($N_{train} = 6K$), but the metrics are computed with respect to the whole Twitter-GloVe *reference*, similar to NeuralGW, RegGW, CycleGW and FlowGW. The colored charts (baselines, whole Twitter-GloVe *reference*) are available in Figure 7 in Appendix B.1. While the comparison of the methods trained on 3K samples (baselines in gray) and 180K samples (our approach) might seem a bit unfair, we stress that the sizes of datasets are selected based on the computational capabilities of the solvers.

Our results are presented in Figure 5. As we can see, NeuralGW is the only method which may deal with large datasets for all correlatedness levels, because it is based on conventional stochastic learning with batches. Even the advanced baselines (RegGW and FlowGW) failed to achieve reasonable performance for $\alpha < 1$ (partially correlated setup). Probably, this is due to inherent reliance on discrete GW techniques. At the same time, our NeuralGW also can demonstrate unsatisfactory quality, see $50 \rightarrow 25$ case in Figure 5 and Figure 8 in the Appendix. Overall, our experiments testify that dimensionality reduction setups are more challenging, which presents an interesting prospective for future research. For the sake of completeness, we additionally provide the GloVe experimental performance for our method trained on $3K$ in Appendix B.1. The results are bad, which is expected because NeuralGW is based on complex adversarial procedure while the dataset is small.

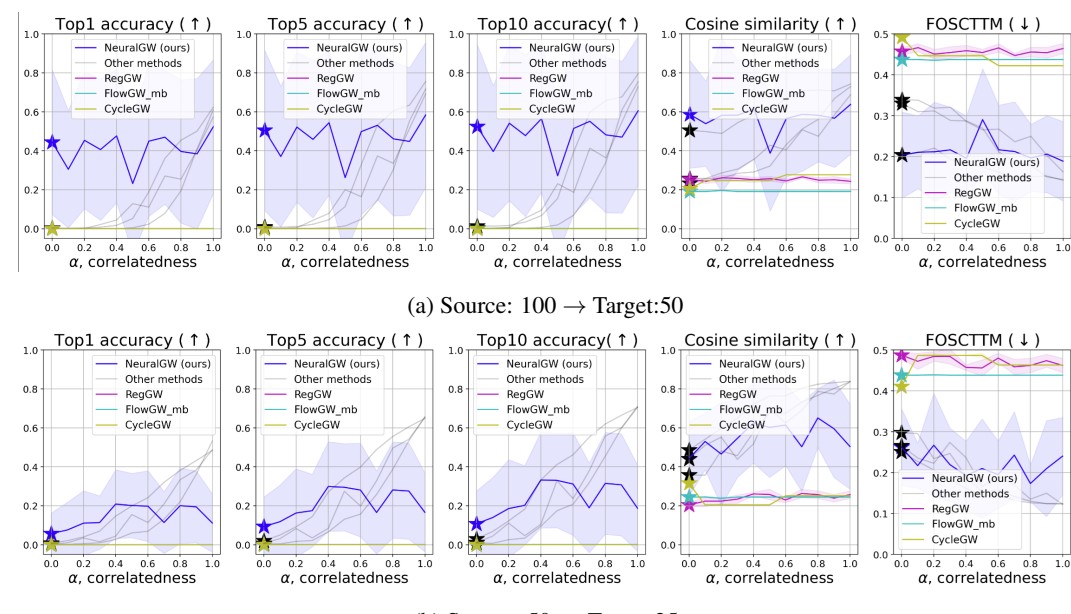

(a) Source: $100 \rightarrow$ Target:50

(b) Source: $50 \rightarrow$ Target:25

Figure 5: Performance of the batched GWOT solvers for the **Twitter-GloVe** embeddings at different levels of correlatedness $\alpha$ in all **high-to-low** setups. The solvers were trained with $N_{train}/2 = 180K$ samples from a whole space of 400K; testing subset consists of $N_{test} = 2048$ samples, the metrics were computed considering the whole 400K samples *reference* space.

**Conclusions.** Our proposed method (NeuralGW) is one of the first solver for the GWOT problem that does not rely on discrete approximations and hence can handle realistic setups with uncorrelated data. Specifically, we attract the readers' attention to metrics' values at $\alpha = 0$ which are highlighted with the star $\star$ symbol. In all the cases (Figure 5), our method outscores competitors by a large barrier. Our NeuralGW supports gradient ascent-descent batch training on large datasets. This capability enables the solver to learn intricate substructures even when trained on uncorrelated data. The initial results for our method suggest the potential to develop a general GWOT solver that is independent of data correlation, a significant advantage given that real-world datasets often lack such correlation.

Despite achieving the best performance on uncorrelated data, the results are inconsistent with respect to the initialization parameters, as evidenced by a high standard deviation among repetitions. This inconsistency may be due to the minimax nature of the optimization problem. Additionally, adversarial methods like our NeuralGW are known to require large amounts of data for training, which can lead to issues when working with small datasets. We explore more general problems for baseline and NeuralGW solvers in Appendix B.

## 6 DISCUSSION

The general scope of our paper is conducting in-depth analyses of machine learning challenges that yield important new insights. In particular, we analyze the sphere of Gromov-Wasserstein Optimal Transport solvers, identify the problems and propose some solutions. Our work clearly shows that while existing Gromov-Wasserstein Optimal Transport methods exhibit considerable success when solving downstream tasks, their performance may severely depend on the intrinsic properties of the training data. We partially address the issue by introducing our novel NeuralGW method. However, it has its own disadvantages. In particular, it is based on adversarial training which may be unstable and is not guaranteed to converge to an optimal solution of the GW problem. Thereby, our work witnesses that GWOT challenge in ML still awaits its hero who will manage to propose a reliable general-purpose method for tackling the problem.

**Reproducibility.** We provide the experimental details in Appendix C and the code to reproduce the results of the conducted experiments in the supplementary material (see `readme.md`).

**Broader impact.** The goal of our paper is to advance the field of ML. There are potential societal consequences of our work, none of which we feel must be specifically highlighted here.

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

## A    PROOFS OF THEOREMS AND LEMMAS.

**Proof of Lemma 5.1.** First, we recall the dual formulation of (1), see, e.g., (Fan et al., 2023):

$$\text{OT}(\mathbb{P}, \mathbb{Q}) \stackrel{\text{def}}{=} \max_f \left[ \min_T \int \big( c(\mathbf{x}, T(\mathbf{x})) - f(T(\mathbf{x})) \big) d\mathbb{P}(\mathbf{x}) + \int f(\mathbf{y}) d\mathbb{Q}(\mathbf{y}) \right], \tag{8}$$

respectively. Note that the existence of a solution $(f^*, T^*)$ of 8 is due to (Fan et al., 2023, Theorem 2). Our proof starts with (5). For each $\mathbf{P}$, we rewrite the inner optimization over $\mathbf{P}$ using (8) for the cost $c(\mathbf{x}, \mathbf{y}) = -\langle \mathbf{Px}, \mathbf{y} \rangle$:

$$\text{Const}\,(\mathbb{P}, \mathbb{Q}) - \text{innerGW}_2^2(\mathbb{P}, \mathbb{Q}) = \max_{\mathbf{P} \in F_{d_x, d_y}} \max_{\pi \in \Pi(\mathbb{P}, \mathbb{Q})} \int_{\mathbb{R}^{d_x} \times \mathbb{R}^{d_y}} \langle \mathbf{Px}, \mathbf{y} \rangle_n \, d\pi(\mathbf{x}, \mathbf{y}) =$$

$$- \min_{\mathbf{P} \in F_{m,n}} \left[ \min_{\pi \in \Pi(\mathbb{P}, \mathbb{Q})} \int_{\mathbb{R}^{d_x} \times \mathbb{R}^{d_y}} -\langle \mathbf{Px}, \mathbf{y} \rangle_{d_y} \, d\pi(\mathbf{x}, \mathbf{y}) \right] =$$

$$- \min_{\mathbf{P} \in F_{d_x, d_y}} \left[ \max_f \int_{\mathbb{R}^{d_x}} f(\mathbf{y}) d\mathbb{Q}(\mathbf{y}) + \min_{T:\, \mathbb{R}^{d_x} \to \mathbb{R}^{d_y}} \int_{\mathbb{R}^{d_x}} -\langle \mathbf{Px}, T(\mathbf{x}) \rangle_{d_y} - f(T(\mathbf{x})) d\mathbb{P}(\mathbf{x}) \right] =$$

$$- \min_{\mathbf{P} \in F_{d_x, d_y}} \max_f \min_T \mathcal{L}(\mathbf{P}, f, T).$$

**Proof of Theorem 5.2**. We expand $\mathcal{L}(\mathbf{P}^*, f^*, T^*)$ and use the fact that $T^*$ is the OT map.

$$\mathcal{L}(\mathbf{P}^*, f^*, T^*) = \int_{\mathbb{R}^{d_y}} f^*(\mathbf{y}) d\mathbb{Q}(\mathbf{y}) - \int_{\mathbb{R}^{d_x}} \langle \mathbf{P}^*\mathbf{x}, T^*(\mathbf{x}) \rangle_{d_y} + f^*\big(T^*(\mathbf{x})\big) d\mathbb{P}(\mathbf{x}). \tag{9}$$

Since $T^*$ is an OT map, we have $T^*_\sharp \mathbb{P} = \mathbb{Q}$, and by the change of variables formula we get:

$$\int_{\mathbb{R}^{d_x}} f^*\big(T^*(\mathbf{x})\big) d\mathbb{P}(\mathbf{x}) = \int_{\mathbb{R}^{d_y}} f^*(\mathbf{y}) d\mathbb{Q}(\mathbf{y}).$$

Plugging this into (9), we get:

$$\mathcal{L}(\mathbf{P}^*, f^*, T^*) = - \int_{\mathbb{R}^{d_x}} \langle \mathbf{P}^*\mathbf{x}, T^*(\mathbf{x}) \rangle_{d_y} d\mathbb{P}(\mathbf{x}).$$

Here, we once again use the fact that $T^*$ is the optimal transport map. Now, since $P^*$ and $f^*$ solve (4), we get the following:

$$\text{innerGW}_2^2(\mathbb{P}, \mathbb{Q}) = \text{Const}(\mathbb{P}, \mathbb{Q}) + \min_{T_\sharp \mathbb{P} = \mathbb{Q}} \mathcal{L}(\mathbf{P}^*, f^*, T)$$

Finally, from the fact that $\pi^* = [\text{id}_{\mathbb{R}^{d_x}}, T^*]_\sharp \mathbb{P}$ is optimal and (9), we have:

$$-\mathcal{L}(\mathbf{P}^*, f^*, T^*) = \int_{\mathbb{R}^{d_x} \times \mathbb{R}^{d_y}} \langle \mathbf{P}^*\mathbf{x}, \mathbf{y} \rangle_{d_y} d\pi^*(\mathbf{x}, \mathbf{y}) = \max_{\pi \in \Pi(\mathbb{P}, \mathbb{Q})} \int_{\mathbb{R}^{d_x} \times \mathbb{R}^{d_y}} \langle \mathbf{P}^*\mathbf{x}, \mathbf{y} \rangle_{d_y} d\pi(\mathbf{x}, \mathbf{y}) =$$

$$- \min_{T_\sharp \mathbb{P} = \mathbb{Q}} \mathcal{L}(\mathbf{P}^*, f^*, T^*),$$

which completes the proof.

## B   EXTENDED EXPERIMENTS

**Overview of the conducted experiments.** To help the reader navigating over all our considered experiments, We provide Table 1 summarizing the full list of experiments in our paper (in the main part of the manuscript and in the appendix).

| Dataset name | Type of embedding | Size of the dataset | Train/test size | Source → Target | Evaluated on |
|---|---|---|---|---|---|
| Twitter | GloVe | 400K | Baseline solvers 6000/2048 | 100→50 | 8048 samples Figure 4a |
| | | | | | 400K samples Figure 7a |
| | | | | 50→25 | 8048 samples Figure 4b |
| | | | | | 400K samples Figure 7b |
| | | | | 50→100 | 8048 samples Figure 6a |
| | | | | | 400K samples Figure 7c |
| | | | | 25→50 | 8048 samples Figure 6b |
| | | | | | 400K samples Figure 7d |
| Twitter | GloVe | 400K | Continuous solvers 360K/2048 | 100→50 | 400K samples Figure 5a |
| | | | | 50→25 | 400K samples Figure 5b |
| | | | | 50→100 | 400K samples Figure 8a |
| | | | | 25→50 | 400K samples Figure 8b |
| Twitter | Byte-Pair | 90K | Baseline solvers 6000/2048 | 100→50 | 90K samples Figure 10 |
| | | | Continuous solvers 88K/2048 | 100→50 | |
| MUSE | Byte-Pair | 90K | Baseline solvers 6000/2048 | 100(English) → 50(English) | 90K samples Figure 9 |
| | | | Continuous solvers 88K/2048 | 100(English) → 50(English) | |
| MUSE | Byte-Pair | 60K | Baseline solvers 6000/2048 | 100(English) → 100(Spanish) | 60K samples Table 3 |
| | | | Continuous solvers 58K/2048 | 100(English) → 100(Spanish) | |

Table 1: Summary of experiments present in the paper.

**Metrics.** We consider three metrics to report: Top $k$-accuracy, FOSCTTM and cosine similarity. In all cases, we require to know the true pairs of the source vectors in the target domain. These true pairs are assumed to be given in some pre-specified *reference* pool of samples, e.g., the full 400K Twitter-GloVe dataset. Under this assumption, the metrics can be defined as follows:

- **Top $k$-accuracy** (↑). Considering the set of $N_{test} = m$ vectors from the source distribution and their predictions. For each predicted vector we compute the $k$-closest (in terms of $L_2$ distance) samples in the *reference* pool of samples and get a sorted set of $k$ indices $\{c_1, c_2, \dots, c_k\}$. As we know the indices of the optimal pairs for the *reference*, we can take the label of the expected optimal pair for any vector in the test source data, this label will be denoted as $l_i$. Therefore, we can define the top $k$-accuracy as follows:

$$\text{Top } k \stackrel{\text{def}}{=} \frac{1}{m} \sum_{j=1}^{m} \mathbb{1}\{l_j \in \{c_1, c_2, \dots, c_k\}\}$$

- **Fraction of Samples Closed Than the True Match (FOSCTTM).** (↓) We calculate the Euclidean distances from a designated transported point ($\mathbf{y} = T(\mathbf{x})$) to every data point from the *reference* set in the target domain. Using these distances, we then compute the proportion of samples that are nearer to the true pair (this information is known). Finally, we take the average of these proportions for all samples. The perfect alignment would mean that every sample is closest to its true counterpart, producing an average FOSCTTM of zero. We note that this metric is rather insensitive to the **reference**, i.e., considering the whole/random subset of Twitter-GloVe dataset does not affect it much.

- **Cosine similarity** (↑). It is computed between the predicted vector and the *reference* (optimal pair) vector in the target space.

### B.1   GloVe

Here we provide additional results for different experimental setups that we considered relevant.

**Low-to-high dimension experiments for baselines.** We consider the combinations 50→100 and 25→50 that were not included in the main text, the metrics were computed as explained in Section §4.2, see Figure 6.

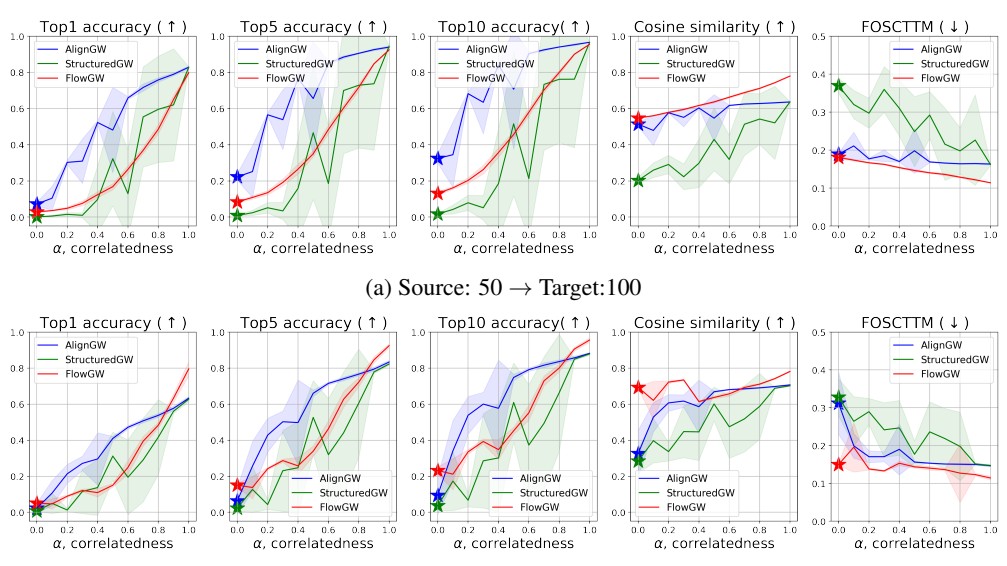

(a) Source: 50 → Target:100

(b) Source: 25 → Target:50

Figure 6: Performance of the baseline GWOT solvers for the **Twitter-GloVe** embeddings at different levels of correlatedness $\alpha$ in **low-to-high** setups. The solvers were trained with $N_{train}/2 = 3000$ samples from a whole space of 400K, this plot shows results for a testing subset of 2048 samples, the metrics were computed considering the $N_{train} + N_{test} = 8048$ samples *reference* space.

**Trained on small dataset, evaluated w.r.t. large *reference*.** The accuracy evaluation involves identifying the $k$-nearest neighbours for a given vector within a target vector space. As discussed in Section §4.2, this space was limited to a small subset of $N_{train} + N_{test}$ samples. This is justified since the methods were trained on a similar number of samples. However, evaluating accuracy across the entire data space could offer deeper insights into the models' ability to capture the intrinsic structures of the probability distributions. These results are presented in Figure 7.

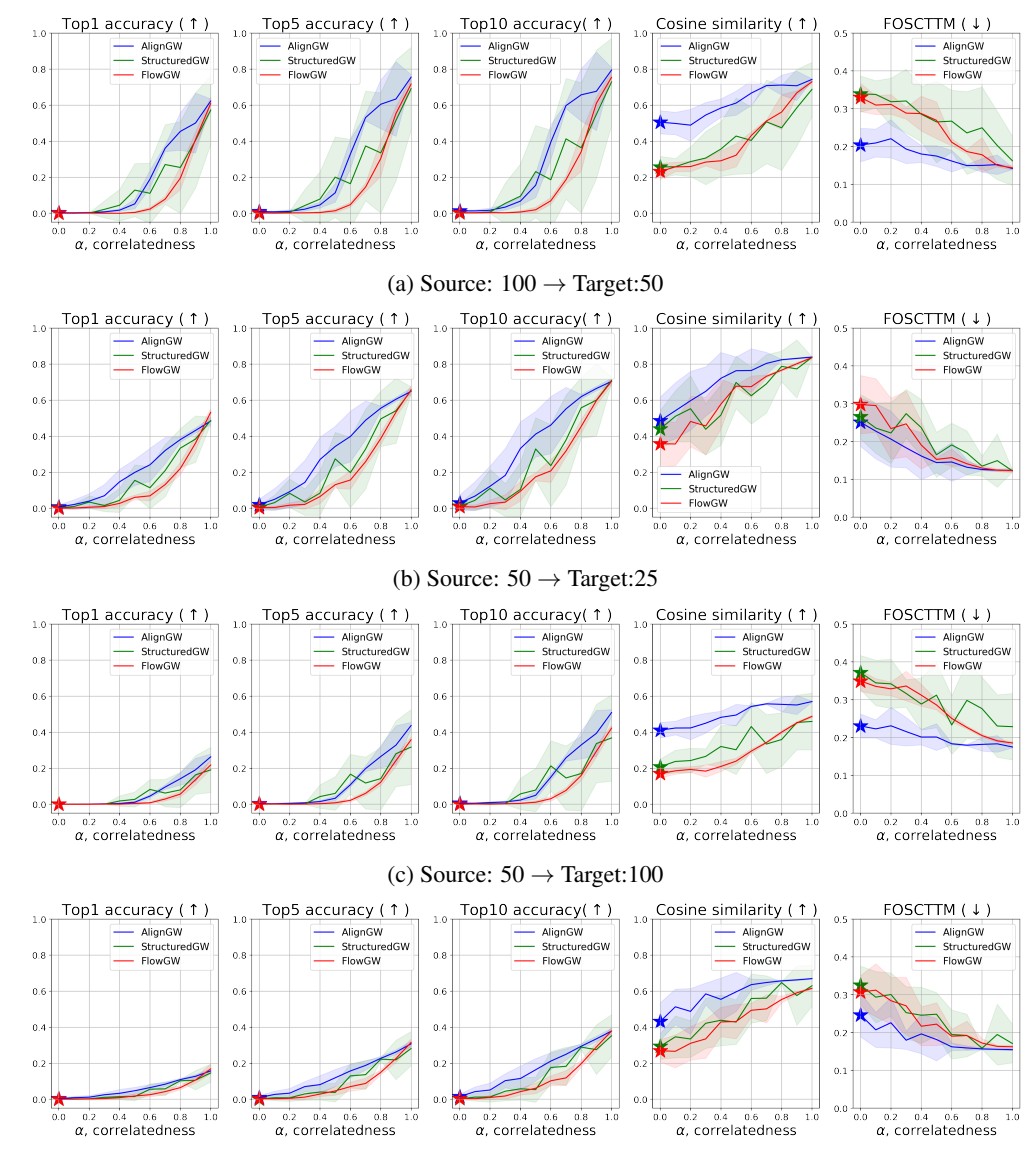

(a) Source: $100 \rightarrow$ Target:50

(b) Source: $50 \rightarrow$ Target:25

(c) Source: $50 \rightarrow$ Target:100

(d) Source: $25 \rightarrow$ Target:50

Figure 7: Performance of the baseline GWOT solvers for the **Twitter-GloVe** embeddings at different levels of correlatedness $\alpha$ in **all** setups. The solvers were trained with $N_{train}/2 = 3000$ samples from a whole space of 400K, this plot shows results for a testing subset of 2048 samples, the metrics were computed considering the whole 400K samples *reference* space.

**Low-to-high experiments for batched solvers.** We consider the combinations $50 \rightarrow 100$ and $25 \rightarrow 50$ for the FlowGW (mini-batch training), RegGw and NeuralGW solvers. One important remark is that for these experiments, only three values of correlatedness were used ($\alpha = 0.2, 0.9, 1.0$) for FlowGW and NeuralGW, the plot can be seen in Figure 8.

**Additional experiments for NeuralGW for high-to-low dimension in small dataset ($N_{train} = 6K$).** Here we consider the same setup used for the baseline methods in §4.2 to show how a neural approach performs when it is trained using a limited amount of samples, $N_{train} = 6K$, see Table 2.

**Conclusion.** For low-to-high experiments, baseline models performance is similar to the high-to-low case showed in §4.2 of the main work. When the trained models are evaluated w.r.t. to the whole data space *reference*, the accuracy drops significantly, from this we can conclude that learning on a small batch does not ensure the inner geometric structure of the space is accurately learned.

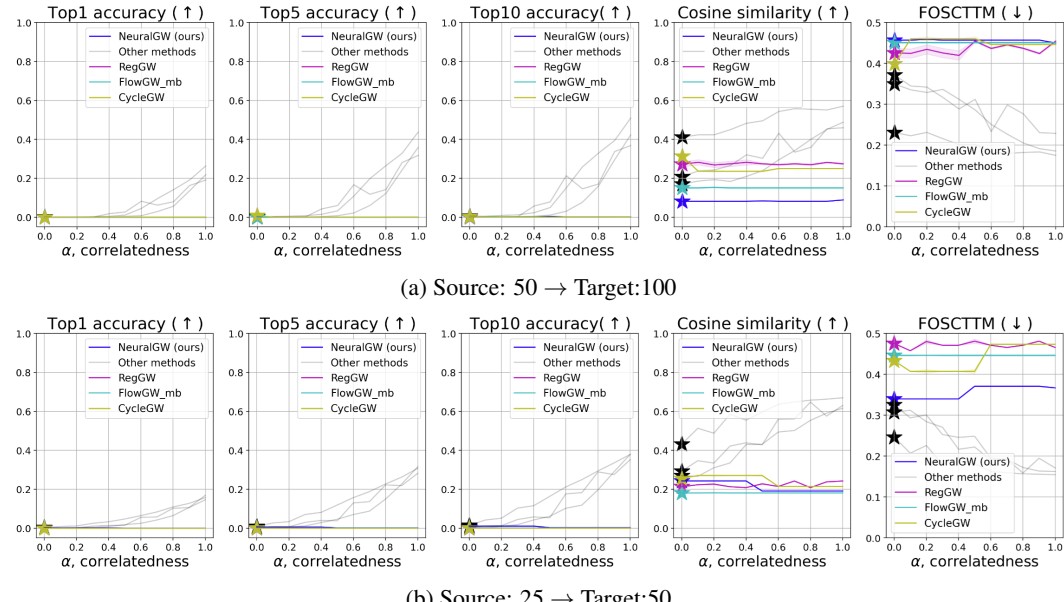

(a) Source: $50 \to$ Target:100

(b) Source: $25 \to$ Target:50

Figure 8: Performance of the batched GWOT solvers for the **Twitter-GloVe** embeddings at different levels of correlatedness $\alpha$ in all **high-to-low** setups. The solvers were trained with $N_{train}/2 = 180K$ samples from a whole space of 400K, this plot shows results for a testing subset of 2048 samples, the metrics were computed considering the whole 400K samples *reference* space.

| Dimensions | Correlatedness | Top 1 | Top 5 | Top 10 | Cosine similarity | FOSCTTM |
|---|---|---|---|---|---|---|
| | $\alpha = 0.2$ | 0.000 | 0.000 | 0.003 | 0.100 | 0.462 |
| $100 \to 50$ | $\alpha = 0.5$ | 0.000 | 0.001 | 0.004 | 0.107 | 0.470 |
| | $\alpha = 1.0$ | 0.000 | 0.003 | 0.005 | 0.117 | 0.447 |
| | $\alpha = 0.2$ | 0.000 | 0.001 | 0.003 | 0.165 | 0.441 |
| $50 \to 25$ | $\alpha = 0.5$ | 0.001 | 0.004 | 0.007 | 0.176 | 0.423 |
| | $\alpha = 1.0$ | 0.000 | 0.000 | 0.004 | 0.174 | 0.435 |
| | $\alpha = 0.2$ | 0.000 | 0.002 | 0.003 | 0.086 | 0.455 |
| $50 \to 100$ | $\alpha = 0.5$ | 0.000 | 0.003 | 0.005 | 0.084 | 0.459 |
| | $\alpha = 1.0$ | 0.000 | 0.002 | 0.003 | 0.089 | 0.450 |
| | $\alpha = 0.2$ | 0.000 | 0.002 | 0.005 | 0.113 | 0.440 |
| $25 \to 50$ | $\alpha = 0.5$ | 0.000 | 0.000 | 0.001 | 0.095 | 0.464 |
| | $\alpha = 1.0$ | 0.001 | 0.003 | 0.005 | 0.124 | 0.436 |

Table 2: Performance of the NeuralGW solver for the **Twitter-GloVe** embeddings at different levels of correlatedness $\alpha$ in **all** setups. The solver was trained with $N_{train}/2 = 3000$ samples from a whole space of 400K, this plot shows results for a testing subset of 2048 samples, the metrics were computed considering the whole 400K samples *reference* space.

NeuralGW cannot properly keep the isometry of the probability space when these are learned from a lower dimension. RegGW and FlowGW (mini-batch) succeed in low-to-high experiments when the source and target spaces are fully correlated.

Finally, NeuralGW is unable to model the inner structures when the number of training samples is small (6K). In general, adversarial algorithms like those used to train NeuralGW require plenty of data to obtain meaningful results.

### B.2 BPEMB

In this section, we explore the motivation behind using Byte-Pair Embeddings, followed by an explanation of how we constructed our new dataset utilizing the MUSE bilingual dictionaries.

The MUSE embeddings were originally obtained using fastText (Joulin et al., 2016); however, most of the methods explored in this paper are unable to align these embeddings effectively. In the work we reference as the AlignGW solver (Alvarez-Melis & Jaakkola, 2018), the authors report positive alignment results on this dataset. However, it is important to note that these results may not fully reflect the true alignment capability of the method, as they incorporate cross-domain similarity local scaling (CSLS) (Conneau et al., 2017). While CSLS is commonly used in alignment tasks to enhance inference performance in multilingual translation, it introduces a corrective bias that may inflate the method's apparent success. This reliance on CSLS may thus lead to results that do not accurately represent the method's intrinsic alignment efficacy. In light of these considerations, we determined that a different approach was necessary for embedding the words from the MUSE vocabularies.

We generated the new dataset using the bpemb library[3], which provides pre-trained subword embeddings. For a thorough explanation of how these embeddings are derived, we recommend reviewing the original paper. To increase the chances that a word from the MUSE dictionaries appear in the BPEmbd vocabularies, we selected the largest available vocabulary size (200K) when loading the pre-trained embeddings. However, if a word still doesn't match, we chose to exclude them. Other reason to consider BPEmb is the possibility to compute the embeddings in different dimensions. For our experiments we only considered English and Spanish as bilingual dictionaries are provided for them. We excluded words with several translations and words with no translations. By doing this we ensure the obtained dataset of source and target embeddings fits our definition of correlatedness in Section §4.1.

With these considerations in mind, we constructed source and target datasets of BP embeddings for MUSE (English and Spanish) and Twitter corpora. Although the number of samples was reduced, the datasets remain viable for continuous methods. We then proceeded with the following experiments:

**MUSE-BPEmb, source: English (100) → target: English (50)** In this experiment we considered the English language for source and target datasets, but the dimension of embedding is different. The total number of samples was $N = 90K$, $N_{train} = 6K$ for baseline solvers and $N_{train} = 87K$ for NeuralGW, $N_{test} = 2048$ for both cases. The metrics were computed using the whole *reference* space, similarly as in the second experiment in Appendix B.1. See Figure 9 for the results.

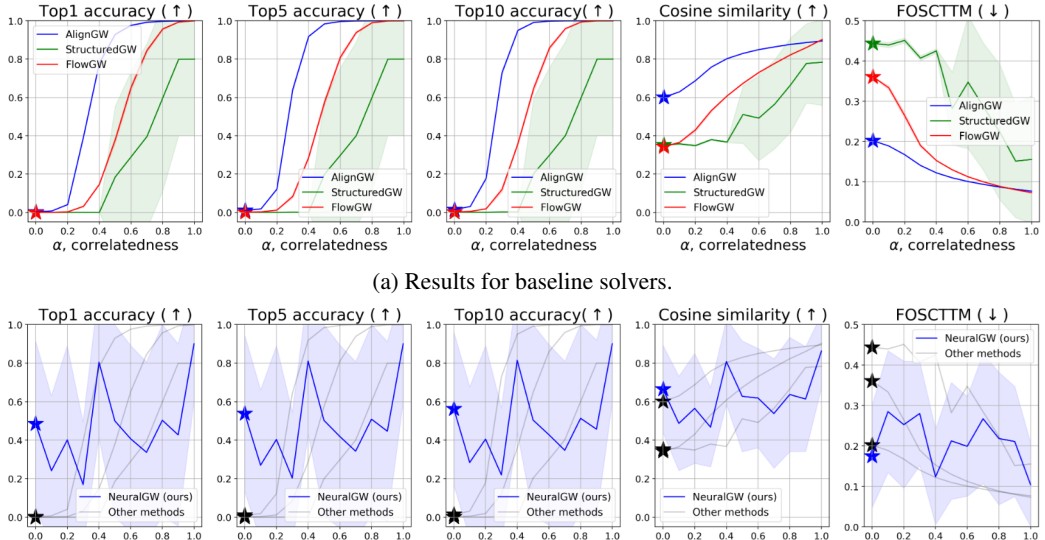

(a) Results for baseline solvers.

(b) Result for NeuralGW.

Figure 9: Performance of the baseline and batched GWOT solvers for the **MUSE-Byte-Pair** embeddings for English language at different levels of correlatedness $\alpha$ in the $100 \to 50$ setup. **(a)** Baseline solvers trained with $N_{train}/2 = 3000$ samples from a whole space of 90K. **(b)** Batched solvers trained with $N_{train}/2 = 43K$ samples. In both cases the plot shows results for a testing subset of 2048 samples, the metrics were computed considering the whole 90K samples *reference* space.

---

[3] https://github.com/bheinzerling/bpemb/tree/master

**MUSE-BPemb, source: English (100) → target: Spanish (100)** We considered two different languages for source and target, the dimension of embedding was equal. The total number of samples was $N = 60K$, $N_{train} = 6K$ for baseline solvers and $N_{train} = 57K$ for NeuralGW, $N_{test} = 2048$ for both cases. The metrics were computed using the whole *reference* space. See Table 3 for the results.

| | **Baseline Solvers** | | | | | |
| --- | --- | --- | --- | --- | --- | --- |
| | **FlowGW** | | **AlignGW** | | **StructuredGW** | |
| $\alpha$ | **Top 10** | **FOSCTTM** | **Top 10** | **FOSCTTM** | **Top 10** | **FOSCTTM** |
| **0.0** | 0.0013 | 0.4222 | 0.0012 | 0.4056 | 0.0000 | 0.5037 |
| **0.5** | 0.0029 | 0.3983 | 0.0022 | 0.3768 | 0.0006 | 0.4541 |
| **1.0** | 0.0267 | 0.3321 | 0.0146 | 0.3085 | 0.0009 | 0.4443 |
| | **Continuous solvers** | | | | | |
| | **RegGW** | | **FlowGW (mb)** | | **NeuralGW (ours)** | |
| $\alpha$ | **Top 10** | **FOSCTTM** | **Top 10** | **FOSCTTM** | **Top 10** mean (std) | **FOSCTTM** mean (std) |
| **0.0** | 0.0001 | 0.4521 | 0.0006 | 0.4767 | 0.0518 (0.0633) | 0.3811 (0.1286) |
| **0.5** | 0.0009 | 0.4411 | 0.0007 | 0.4770 | 0.0284 (0.0564) | 0.4379 (0.1067) |
| **1.0** | 0.0013 | 0.4384 | 0.0008 | 0.4772 | 0.0748 (0.0751) | 0.3514 (0.1321) |

Table 3: Results for MUSE dataset for English and Spanish as source and target languages, respectively, both are 100-dimensional BP embeddings.

**Twitter dataset with different dimension of embeddings:** For this case we considered the same dataset as for the GloVe experiments, but we changed the type of embedding to BPEmb, only the experiment for source: $100 \rightarrow$ target: 50 was performed. The total number of samples was $N = 90K$, $N_{train} = 6K$ for baseline solvers and $N_{train} = 87K$ for NeuralGW, $N_{test} = 2048$ for both cases. The metrics were computed as in the previous experiment. See Figure 10 for the results.

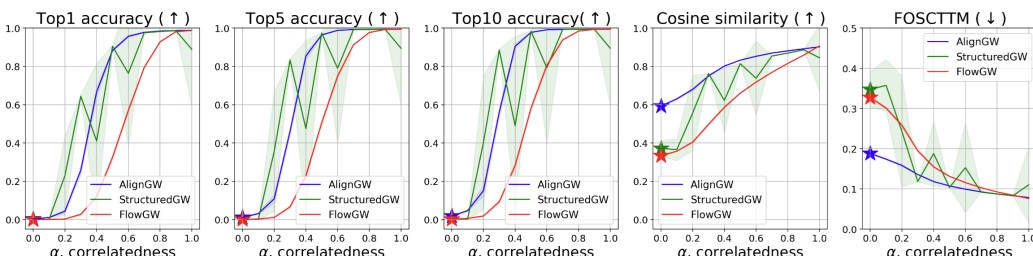

(a) Results for baseline solvers.

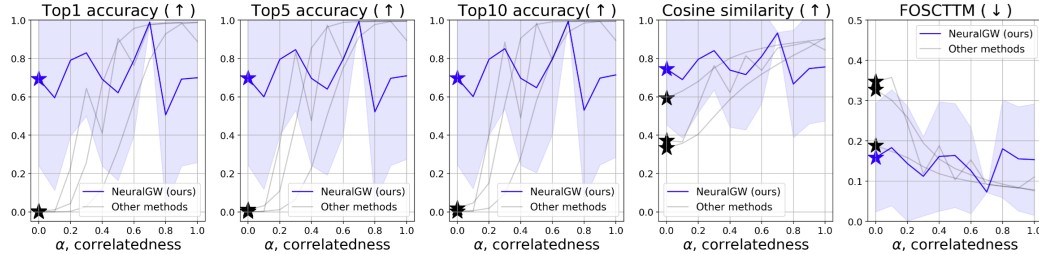

(b) Result for NeuralGW.

Figure 10: Performance of the baseline and batched GWOT solvers for the **Twitter-Byte-Pair** embeddings at different levels of correlatedness $\alpha$ in the $100 \rightarrow 50$ setup. **(a)** Baseline solvers trained with $N_{train}/2 = 3000$ samples from a whole space of 90K. **(b)** Batched solvers trained with $N_{train}/2 = 43K$ samples. In both cases the plot shows results for a testing subset of 2048 samples, the metrics were computed considering the whole 90K samples *reference* space.

For the sake of clarity, we only computed the metrics in the whole target space as in the second additional experiment in Appendix B.1. The number of training samples was kept $N_{train} = 6K$ for baseline solvers and $N_{train} = 87K$ for NeuralGW, $N_{test} = 2048$ for both cases.

**Conclusions.** The obtained results with BPEmb embeddings clearly demonstrate the same trends as for GloVe embeddings (Sections 4.2 and 5.2 of the main text). For both MUSE and Twitter datasets, our NeuralGW turns out to be the only method capable to leverage uncorrelated ($\alpha = 0$) setup when learning the map between embeddings of the same language but with different dimensions. At the same time, dealing with more complex *iter*-language setup turns out to be more difficult and none of the solvers have reached satisfactory quality on this setting. This underlines the inherent complexity of the GW problem.

### B.3 BIOLOGICAL DATASET

As seen in the previous experiments, the baselines and NeuralGW solvers have their own limitations and drawbacks heavily linked to their nature. However, in spite of their independent performance, they could partially recover the inner geometry of the domains. In this section, we propose a stress test scenario in which the solvers of the conventional GWOT problem yield poor results.

We explore the performance of the baselines solvers and NeuralGW in a biological dataset called bone marrow (Luecken et al., 2021) which is considered in the FlowGW paper (Klein et al., 2023). This dataset consists of 6224 samples of two different RNA profiling methods (ATAC+GEX and ADT+GEX), the samples in each domain belong to the same donors, therefore, the real pairs are known. The dimensionality for source and target domains are 38 and 50, respectively. Results can be found in Table 4. All the solvers are tested using 5000 samples for training and the rest for testing.

| | **FlowGW** | | **AlignGW** | | **StructuredGW** | | **NeuralGW** | |
|---|---|---|---|---|---|---|---|---|
| $\alpha$ | Top 10 | FOSCTTM | Top 10 | FOSCTTM | Top 10 | FOSCTTM | Top 10 | FOSCTTM |
| **0.2** | 0.004 | 0.486 | 0.004 | 0.488 | 0.001 | 0.489 | 0.003 | 0.479 |
| **0.5** | 0.004 | 0.489 | 0.004 | 0.483 | 0.004 | 0.487 | 0.003 | 0.514 |
| **1.0** | 0.004 | 0.49 | 0.004 | 0.484 | 0.004 | 0.49 | 0.003 | 0.459 |

Table 4: Results for bone marrow dataset.

**Conclusions.** In all the cases, the solvers could not properly replicate the inner geometry of the distributions even for totally correlated setups, this leaded to get metrics corresponding to random guessing, i.e. accuracies close to 0 and FOSCTTM close to 0.5. There is one case of success for a solver trained on this dataset which corresponds to FlowGW (Klein et al., 2023), however, the reported results in their paper were obtained using a fused-GW solver instead of a conventional GW.

Finally, we can state that there is no general solver for the GWOT problem, all the currently available methods struggle when dealing with real world scenarios, i.e. uncorrelated data, or with real world datasets, i.e. not consistent inner structures.

## C SOLVERS' IMPLEMENTATION DETAILS

All the experiments were done without any normalization for the source and target vectors and for all the studied methods (baselines and NeuralGW). A total of ten repetitions were performed. It is important to clarify that the parameters listed below are the ones we used to align the embeddings, they may require some tweaks to make them work in the toy setup.

**StructuredGW.(Sebbouh et al., 2024)** We used the code from the official repository:

```
https://github.com/othmanesebbouh/prox_rot_aistats
```

As specified in Section §3, the algorithm uses an iterative solver that updates the cost matrix **T** by implementing several methods depending on the type of regularization, we only use the exact computation without any regularization. The plan $\pi$ is also updated every iteration by performing Sinkhorn iterations, we set this number of iterations to 1000. The entropy is set to $\varepsilon = 1e\text{-}4$. The total number of iterations is set to 200 or until convergence.

In this implementation, the authors use the Optimal Transport Tools (OTT) library (Cuturi et al., 2022). The computation time per repetition until convergence was 30 minutes in average on a CPU.

**AlignGW.(Alvarez-Melis & Jaakkola, 2018)** We use the official implementation of the method taken from the repository:

https://github.com/dmelis/otalign

We set the entropy term to $\varepsilon =$1e-4 and use the cosine similarity to compute the source and target intra-cost matrices $\mathbf{C}^x$ and $\mathbf{C}^y$. We later normalize them by dividing them by their respective means as proposed in the original implementation. The model was trained on a CPU and the average training time was 30 minutes per repetition. The implementation uses the POT library. We train a scikit-learn's MLPRegressor on top of it as an inference method for test data, the parameters are: hidden_layer_sizes=256, random_state=1, max_iter=500.

**FlowGW. (Klein et al., 2023)** We used the implementation provided in the OTT library for the GENOT with slight modifications to adapt it to our pipeline. The hyperparameters for the vector field were as follows: Number of frequencies: 128, layers per block: 8, hidden dimension: 1024, activation function: SiLu, optimizer: AdamW (lr=1e-4). The Gromov-Wasserstein solver was set to work with entropy $\varepsilon = 1e - 3$ and using cosine similarity distance to compute the intra-domain matrices.

**RegGW. (Uscidda et al., 2024)** For the sake of fairness, our implementation of this solver is based on the publicly available implementation for the Monge gap regularizer from the OTT library (Cuturi et al., 2022). To compute the Gromov-Wasserstein distance we used the GW solver from the library and took the entropy regularized cost. The following parameters were used for training: $\varepsilon_{fit} = 0.01$, $\varepsilon_{reg} = 0.001$, $\lambda = 1$. The transport model was parametrized as an MLP with $[512, 256, 256]$ dimensions for the hidden layers, the optimizer learning rate was 1e-4 and a batch size of 256.

**CycleGW (Zhang et al., 2021)**. For the implementation of this solver, we used the code provided by the authors in their repository:

https://github.com/ZhengxinZh/GMMD

As in the original implementation, we used fully connected neural networks (FCNN) to parametrize $f$ and $g$, in both cases the network consisted on a single layer with 512 neurons, and trained using the Adam optimizer with a learning rate of $1e$-3 (as suggested in the original paper). Both regularization parameters, $\lambda_x$ and $\lambda_y$ in the original paper, were set to $0.1$. The multiplier of the distortion term was set to $5e$-4. In spite of following the original implementation, it was not possible to make the solver work for our setups beyond the toy experiment.

C.1 NEURALGW.

The innerGW problem in (4) can be optimized using our Algorithm 1.

---

**Algorithm 1:** Training algorithm for Neural Gromov-Wasserstein OT

1 **Input:**Distributions $\mathbb{P}$ and $\mathbb{Q}$ obtained from samples.
2 **Output:**Optimal rotation matrix $P_\omega$, critic $f_\theta$ and transport map $T_\gamma$.
3 **for** $i = 1, 2, 3, \ldots, n_{epochs}$ **do**
4     Sample batch from source and target distributions $X \sim \mathbb{P}, Y \sim \mathbb{Q}$.
5     **for** $i = 1, 2, 3, \ldots, k_P$ **do**
6         Compute $P$ loss $\mathcal{L}_P = -\frac{1}{N}\sum_{n=1}^{N}\langle\mathbf{P}_\omega\mathbf{x}, T_\gamma(\mathbf{x})\rangle$
7         Gradient step over $\omega$ using $\frac{\partial\mathcal{L}_P}{\partial\omega}$
8         **for** $j = 1, 2, 3, \ldots, k_f$ **do**
9             **for** $k = 1, 2, 3, \ldots, k_T$ **do**
10                 Compute mover loss $\mathcal{L}_T = -\frac{1}{N}\sum_{n=1}^{N}\langle\mathbf{P}_\omega\mathbf{x}, T_\gamma(\mathbf{x})\rangle - \frac{1}{N}\sum_{n=1}^{N}f_\theta(T(\mathbf{x}))$
11                 Gradient step over $\gamma$ using $\frac{\partial\mathcal{L}_T}{\partial\gamma}$
12             Compute critic loss $\mathcal{L}_f = -\frac{1}{N}\sum_{n=1}^{N}f_\theta(T_\gamma(\mathbf{x})) - \frac{1}{N}\sum_{n=1}^{N}f_\theta(\mathbf{y})$
13             Gradient step over $\theta$ using $\frac{\partial\mathcal{L}_c}{\partial\theta}$

---

Every experiment runs for 200 epochs. Each epoch iterates over the whole dataset (400K or 6K samples). $f$ and $T$ are parametrized using multi-layer perceptrons with $n_l$ with width $h$, $\mathbf{P}$ is

taken from the matrix of weights of a linear layer, these models are trained for $k_f, k_T$ and $k_p$ iterations, respectively. The implementation details can be seen in Table 5. The batch size is 512 for the experiments with 400K samples and 64 for the experiments with 6K. Our code is written in `PyTorch`.

Table 5: Parameters for NeuralGW.

| | Model | $k$ | $n_l$ | $h$ | lr |
|---|---|---|---|---|---|
| $100{\to}50$ | $f$ | $k_f = 1$ | | | |
| $50{\to}25$ | $T$ | $k_T = 10$ | 4 | 512 | 1e-4 |
| $25{\to}50$ | $P$ | $k_P = 1$ | | | |
| | $f$ | $k_f = 1$ | | | |
| $50{\to}25$ | $T$ | $k_T = 10$ | 4 | 256 | 1e-4 |
| | $P$ | $k_P = 1$ | | | |

Every epoch takes around to 3 minutes running on a GPU NVIDIA Tesla V100.

# D    BROADER DISCUSSIONS

## D.1    DISCRETE GW SOLVERS UNDER LOW CORRELATEDNESS DATA SCENARIO

Our experimental results (Section 4.2 of the main text) testify that the discrete baseline solvers perform unsatisfactory when the data correlatedness level $\alpha$ tends to zero. We hypothesis that the main reason behind this behavior is as follows. Small amount of data used for discrete solvers hardly could "catch" the intrinsic geometry of the underlining distribution. When we apply discrete GW solver, the Gromov-Wasserstein mapping is learned between the geometries induced by sample distributions, not original distributions, see Figure 11 for illustration. These "induced" geometries may be different from the original ones, they may have other symmetries and other properties. Matching them may result in Gromov-Wasserstein map which is quite different from the real GW map.

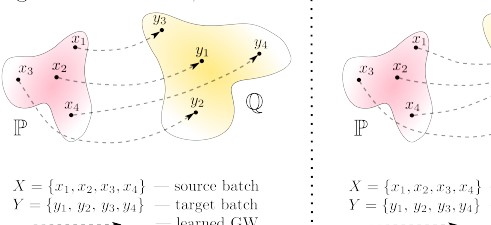

Figure 11: Discrete GW maps fitted under high (left) and low (right) correlatedness level.

On the other hand, when correlatedness level is high ($\alpha = 1$), the GW problem is reduced to finding the proper permutation of the data, see Figure 11, left part. The true solution of discrete GW in this case *coincides* with the true underlining GW map. If the learned map properly generalizes to new (unseen) samples, then the resulting performance is expected to be satisfactory. It is the behaviour we observe in our experiments, Section 4.2.

