# OpenReview forum: "Uncovering Challenges of Solving the Continuous Gromov-Wasserstein Problem"
_ICLR.cc/2025/Conference — Submitted to ICLR 2025_

### Official Review · Reviewer_4w5B · 2024-10-28

**Soundness:** 3
**Presentation:** 3
**Contribution:** 2
**Rating:** 6
**Confidence:** 4

**Summary:**

This paper is an attempt to answer the promising question of solving continuous Gromov-Wasserstein Optimal Transport (GWOT) problem. The authors crash-test existing continuous GWOT approaches on different scenarios, carefully record and analyze the obtained results, and identify issues. Their findings experimentally testify that the scientific community is still missing a reliable continuous GWOT
solver, which necessitates further research efforts. As the first step in this direction, the authors propose a neural continuous GWOT method which does not rely on discrete techniques. Their method is based on the key observation that the reformulation of dual continuous GWOT problem can be solved as a min-max-min optimization problem. By simply parameterization with neural networks, the authors use the alternating stochastic gradient ascent/descent/ascent method to train their parameters. The empirical performance is encouraging.

**Strengths:**

The paper conducts a deep analysis of existing papers and reveal that one important characteristic that may greatly affect practical performance is the considered data setup. By following these findings, the authors evaluate the performance of existing continuous GWOT solvers in more statistically fair and practically realistic uncorrelated data setups. To alleviate the dependence on the mutual statistical characteristics of the source and target training data, a continuous neural GW solver was proposed and performed well in practice.

**Weaknesses:**

1. The findings of this paper are interesting and reveal that the empirical success of the existing GWOT solvers seems to be a bit
over-estimated and requires to be treated more critically. However, the interpretation is not thorough. Does this limitation only come from that all existing continuous GWOT solvers heavily rely on discrete techniques?

2. Your continuous neural GW solver is insightful. Could you provide some theoretical guarantee? I do not think the complexity analysis is necessary but at least your method needs to be guaranteed to converge under certain condition.

**Questions:**

Please see the weakness part.

---

> ### Author Response · Authors · 2024-11-22
> **Answer to reviewer 4w5B**
>
> Dear reviewer, thanks for your feedback and for the positive evaluation of our work. Below we address your questions and comments. A note about the modifications we made in the revised version of our manuscript as well as supplementary materials (code) is available in our general "Modifications" [comment](https://openreview.net/forum?id=sRaAt9OOnW&noteId=PZrOwCkVpG).
>
> ## [Weakness 1:] The findings of this paper are interesting and reveal that the empirical success of the existing GWOT solvers seems to be a bit over-estimated and requires to be treated more critically. However, the interpretation is not thorough. Does this limitation only come from that all existing continuous GWOT solvers heavily rely on discrete techniques?
>
> We cannot claim that the limitations of these solvers arise solely from the discrete techniques they employ, as our own solver, NeuralGW, is also imperfect and heavily depends on its initialization parameters. The only we can claim - is that the reliance on discrete techniques may (and seems to ) cause the poor performance - we added an additional discussion about the issue of discrete techniques under uncorrelated setup in Appendix D.1.
>
> We think, the best characterization of the situation is that the challenges faced by all the continuous solvers stem directly from the inherent difficulty of solving the Gromov-Wasserstein problem.
>
> ## [Weakness 2:] Your continuous neural GW solver is insightful. Could you provide some theoretical guarantee? I do not think the complexity analysis is necessary but at least your method needs to be guaranteed to converge under certain condition.
>
> We thank the reviewer for posting this question. However, we think that providing a good answer to this question is not that easy, and hypothetical convergence guarantees for our algorithm (if they are possible) may turn out to be too restrictive, and, therefore, practically not very useful.
>
> We have two arguments to support our claim.
>
> - The first argument is the complexity of GW problem itself. Under certain (and rather mild) condition, the existence of optimal GW map holds, but it may be non-unique. For example, in the simplest Gaussian-to-Gaussian case (where the second Gaussian is zero centered), if a map $T^*$ is an optimal GW map, so does the composition of $T^*$ with any rotation. For the more formal and extended explanation, please see (Vayer et. al., 2020, Example 4.2.1). It is not clear how our NeuralGW solver behaves under the presence of such symmetries. Probably, some regularizations of the algorithm could help to avoid such ambiguities, but theoretical analysis of such regularizations will only make the question harder.
>
> - The second argument relates more to the three-level adversarial objective of our algorithm. From the analysis of related works devoted to adversarial generative models based on Optimal Transport we know that the researches typically establish the convergence guarantees with help of quality bounds analysis, see, inter alia, (Makkuva et. al., 2020, Theorem 3.6); (Rout et. al., 2022, Theorem 4.3); (Fan et. al., 2023, Theorem 4). The general idea behind all of these theorems is to restrict the error of recovered (OT) map given the errors at each level of adversariality. Compared to all of the aforementined works which have *two* levels of adversariality, our NeuralGW has *three* levels, which much complicates the analysis. Moreover, the quality bounds analysis typically places too restrictive assumptions on the solution (e.g., requires *strong* convexity of dual Kantorivich potential $f$ - we also have this potential in our objective (6)), which hardly could be satisfied (and even checked) in practice.
>
> Based on this, we humbly think that conducting the convergence analysis of our NeuralGW is rather a good direction of further theoretical research. We underline that the primary focus of our paper is to establish the critical gap in the existing methodology for validating (continuous) GW solvers.
>
>
> ### Concluding remarks.
>
> Please reply to our post and inform us if the clarifications provided adequately address your concerns regarding our work. We are more than willing to discuss any remaining points during the discussion phase. If the responses offered meet your satisfaction, we kindly request you to consider raising your score.
>
> **References**
>
> Vayer, Titouan. "A contribution to optimal transport on incomparable spaces." arXiv preprint arXiv:2011.04447 (2020).
>
> Makkuva, Ashok, et al. "Optimal transport mapping via input convex neural networks." International Conference on Machine Learning. PMLR, 2020.
>
> Rout, Litu, Alexander Korotin, and Evgeny Burnaev. "Generative Modeling with Optimal Transport Maps." International Conference on Learning Representations. 2022.
>
> Fan, Jiaojiao, et al. "Neural Monge Map estimation and its applications." Transactions on Machine Learning Research. 2023.

---

> > ### Author Response · Authors · 2024-12-02
> > **Additional comment for Reviewer 4w5B**
> >
> > Dear Reviewer,
> >
> > The deadline for the rebuttal phase is approaching. We would appreciate your feedback on our responses to the reviews. We are happy to address any additional questions during the remaining period.

---

### Official Review · Reviewer_ctGz · 2024-11-03

**Soundness:** 4
**Presentation:** 3
**Contribution:** 3
**Rating:** 8
**Confidence:** 4

**Summary:**

The paper considers the continuous GWOT problem and presents a min-max optimization approach as a solution.
The authors focus on two main problems in the paper:
- establishing deficiencies in the existing solvers. In particular, the focus of this discussion is on the quality (or correlatedness) of training samples.
- proposing a solver that addresses the above-mentioned deficiencies by utilizing the smoothness of the underlying distributions.

The paper is well-written and clear. Moreover, the authors established the motivation for the problem and the scope of the study well. The results presented within are clear and concise.

Having said that, I have the following comments.
- This work seems to be one of the first to address this problem in a continuous setting. Thus, comparative studies are difficult, and hence, the significance of the work is hard to gauge.
- Moreover, the proposed method performs well in some aspects and poorly in others when compared to existing solvers. Hence, although the insight into the drawbacks of other solvers and the subsequent solution are interesting, I am uncertain if the results are important enough to recommend an acceptance strongly.

**Strengths:**

- The results seem consistent (or unaffected) by the correlatedness of the source and target samples. This is a significant advantage as the data preprocessing burden is reduced.
- Moreover, this method is proposed for continuous GWOT problems, which other solvers currently do not handle.

**Weaknesses:**

Based on my understanding, the following could be considered limitations of this approach:
- Since the method is exclusively for the continuous GWOT problem, the approach presented here would not be helpful in applications where a discrete measure makes better sense.
- As the authors themselves identified, the method requires large datasets for training, and in general, the results seem to be sensitive to optimization parameters and training data variation.

**Questions:**

- I think the Toy 3D->2D problem section needs better explanation. I was quite lost when trying to understand the objective of the experiment or what was being done. Were the mixed Gaussian distributions in the 3D and 2D spaces the same? What do the colors in the plot represent? It might be better to move this part to the appendix because this step merely seems to be a way to validate the implementation of various solvers.
- Not much discussion or insight is present into the reason behind the performance seen in the simulation. What is the reason behind the better performance of NeuralGW when the source and target samples are uncorrelated? Does the smoothness of the underlying distributions affect the variance seen in repetitions?
- Is there some way to quantify the algorithm's sensitivity to the optimization parameters?

---

> ### Author Response · Authors · 2024-11-22
> **Answer to reviewer ctGz**
>
> Dear reviewer, thanks for your feedback and for the highly positive evaluation of our work. Below we address your questions and comments. A note about the modifications we made in the revised version of our manuscript as well as supplementary materials (code) is available in our general "Modifications" [comment](https://openreview.net/forum?id=sRaAt9OOnW&noteId=PZrOwCkVpG).
>
> ## [Weakness 1:] Since the method is exclusively for the continuous GWOT problem, the approach presented here would not be helpful in applications where a discrete measure makes better sense.
>
> Even though we agree on the comment, we do not see this point as a weakness of our solver since discrete and continuous solvers are effective for applications of different spheres, i.e. discrete solvers can be used when the data is scarce like in biological applications while a continuous solver is more effective when large datasets can be accessed like in the case of word embeddings.
>
> ## [Weakness 2:] As the authors themselves identified, the method requires large datasets for training, and in general, the results seem to be sensitive to optimization parameters and training data variation.
>
> This limitation is indeed does exist, and the reason is that our NeuralGW uses several neural networks and adversarial training. However, we want to emphasize that the same limitation is, in principle, applicable to any adversarial generative model (GANs), which also typically requires large amount of training data. The other possible source of non-robustness of our algorithm to the hyperparameters is the complexity of GW problem itself. In particular, GW problem between some source and target domains may have *multiple* solutions (e.g. between Gaussian distributions), which is another source of headache for our training procedure.
>
> Also, it is relevant to highlight that the mentioned restriction can be alleviated by using a more friendly type of embedding technique, this can be seen in our new experimental results (please see second point of our [general response](https://openreview.net/forum?id=sRaAt9OOnW&noteId=wdEmixZ12I) for the details) where we used the Byte-Pair embeddings (BPEmb). More specifically, we refer to our first (MUSE dataset, BPEmb embedding) and third  (Twitter dataset, BPEmb) additional experiments. For the results, please see Figures 9 and 10 in the revised version of our manuscript, which demonstrate satisfactory performance of our solver. In these experiments with BPEmb, the considered dataset sizes (90K) are considerably smaller compared to our original experiments with GloVe embedder (360K). Note that the actual training number of samples for BPEmb was 90K / 2 = 45K.
> As the conclusion, we can infer that NeuralGW can actually work with smaller datasets when the type of embedding offers more suitable structures for GW solvers.
>
> ## [Question 1:] I think the Toy 3D->2D problem section needs better explanation. I was quite lost when trying to understand the objective of the experiment or what was being done. Were the mixed Gaussian distributions in the 3D and 2D spaces the same? What do the colors in the plot represent? It might be better to move this part to the appendix because this step merely seems to be a way to validate the implementation of various solvers.
>
> In this experiment, we match two independently generated mixtures of Gaussians, meaning there is no direct relationship between the clusters of the source and target distributions. The colors indicate the source clusters to which each sample belongs. So, the machinery in this experiment is as follows:
>
> - We fit a GW solver to match two unpaired mixtures of gaussians (3D mixture and 2D mixture).
>
> - We assign each cluster in the source distribution a color, and see how the points of each source cluster are transformed (with the fitted GW solver) to the target clusters.
>
> - Since we solve GW problem, we expect a properly fitted GW to map each source cluster/neighbouring clusters of the target distribution. It could be nicely depicted with the spread of colors among the target clusters.
>
> We chose to use colors because maintaining the distances between samples within the same cluster is crucial, and coloring effectively illustrates this relationship.
>
> Even though this experiment is mainly oriented to show the correct implementation of the solvers, we consider it important to properly introduce the performance of the solvers.

---

> ### Author Response · Authors · 2024-11-22
> **Answer to reviewer ctGz. Part 2**
>
> ## [Question 2:] Not much discussion or insight is present into the reason behind the performance seen in the simulation. What is the reason behind the better performance of NeuralGW when the source and target samples are uncorrelated? Does the smoothness of the underlying distributions affect the variance seen in repetitions?
>
> The strong performance of NeuralGW in uncorrelated setups can be attributed to its independence from discrete solvers. For instance, RegGW relies on the entropic regularization cost computed using a discrete solver, while FlowGW is constructed based on a discrete solution. In turn, the poor performance of discrete solvers is discussed in the "Conclusions" paragraph of section 4.2; also we added an extended discussion on this phenomenon in Appendix D.1 of our revised text.
>
> From our recent results (please, take a look at the second point of our [general response](https://openreview.net/forum?id=sRaAt9OOnW&noteId=wdEmixZ12I)), we can conclude that the variance in performance is primarily driven by the initialization of the networks, whereas the type of embedding can influence the overall effectiveness of the solvers.
>
> ## [Question 3:] Is there some way to quantify the algorithm's sensitivity to the optimization parameters?
>
> Although it is possible to quantify the sensitivity by running several iterations with different parameters, it might demand a lot of time due to the amount of parameters and setups we consider in the paper. That is why in our text we only report stds w.r.t. random seeds. We aim to improve the stability of the NeuralGW solver in future works.
>
> ### Concluding remarks.
>
> Please reply to our post and inform us if the clarifications provided adequately address your concerns regarding our work. We are more than willing to discuss any remaining points during the discussion phase.

---

> > ### Comment · Reviewer_ctGz · 2024-11-24
> >
> > I thank the authors for their thorough and detailed responses. I have no further comments.

---

### Official Review · Reviewer_hY8K · 2024-11-04

**Soundness:** 2
**Presentation:** 2
**Contribution:** 2
**Rating:** 6
**Confidence:** 3

**Summary:**

The paper proposes to study the continuous Gromov-Wasserstein (GW) problem, i.e. focusing on the scenario when discrete GW solver may not be scalable or not generate to unseen data. Specifically the paper discusses the applicability of GW to various tasks where a latently matched dataset is in place or not, resulting in different performance. The paper then proposes an algorithm with min/max/min formulation that solves the GWOT problem that is compatible with large scale data and does not rely on discrete GW solvers, thus mitigating the aforementioned issues.

**Strengths:**

The paper points out some issues that have been largely dismissed in the OT and GW community. First the discrete GW solver, when only having access to small batches of datasets that does not have a priori good matching but does have one on a large scale, could have inhomogeneous performance and could be hard to generalize (one such existing attempt is unbalanced GW). Another insight is on the apparent scalability of typical GW solvers, which makes the application of neural networks convincing. The experiments seem thorough, comparing several existing computational method of GW.

**Weaknesses:**

1. Soundness of the theoretical claims:
A major concern is on the main theorems (5.1, 5.2). These results use standard duality and reformulations that involve the optimal Monge map, but the existence is never justified. In particular, it is well known that GW or inner product GW does not necessarily have optimal map, i.e. the minimum may not be achieved [1]. Same for the existence of the optimal potential, see [2]. Thus the claims mostly only serve as heuristics, as theorem 5.2 is solely on characterization of the optimal map. More need to be discussion regarding why parametrizing by neural network is valid when the optimum may not be achieved.


2. Discussion of existing methods and novelty of the proposed algorithm:
The paper claims to be the first in a comprehensive discussion of continuous GW, and the proposed statistical correlatedness is novel. However the paper is missing many discussions on related works. Most of the paper's missing claims on existence and regularity of Gromov-Monge map have been studied in [1],[3],[4]. The statistics question in section 1 " transition from the discrete to the continuous setup may be questionable from a statistical point of view" is discussed in [2]. Neural network based GW solvers have appeared in [5],[6], both claiming similar applicability to large scale data as mentioned in this paper. The existence of a discrete Gromov-Monge map is first shown in [7]. Detailed discussions on related methods should be added to further strengthen the proposed benchmark.


3. A minor comment on the presentation of Section 4: the paper made several claims about discrete GW problems, ranging from practical statistical assumptions to computational difficulties, but it is not entirely lucid what the main issue is: is it the learned map through extrapolation not generalizing, or the sample complexity is bad, or the optimization is harder with uncorrelated data? The argument could be strengthened with discussions on its relation to other classical learning concepts like generalization error, statistical sample complexity, optimization complexity etc.


[1] Dumont, Théo, Théo Lacombe, and François-Xavier Vialard. "On the existence of Monge maps for the Gromov–Wasserstein problem." Foundations of Computational Mathematics (2024): 1-48.

[2] Zhang, Zhengxin, et al. "Gromov–Wasserstein distances: Entropic regularization, duality and sample complexity." The Annals of Statistics 52.4 (2024): 1616-1645.

[3] Mémoli, Facundo, and Tom Needham. "Comparison results for Gromov–Wasserstein and Gromov–Monge distances." ESAIM: Control, Optimisation and Calculus of Variations 30 (2024): 78.

[4] Mémoli, Facundo, and Tom Needham. "Distance distributions and inverse problems for metric measure spaces." Studies in Applied Mathematics 149.4 (2022): 943-1001.

[5] Hur, YoonHaeng, Wenxuan Guo, and Tengyuan Liang. "Reversible Gromov–Monge sampler for simulation-based inference." SIAM Journal on Mathematics of Data Science 6.2 (2024): 283-310.

[6] Zhang, Zhengxin, et al. "Cycle consistent probability divergences across different spaces." International Conference on Artificial Intelligence and Statistics. PMLR, 2022.

[7] Titouan, Vayer, et al. "Sliced gromov-wasserstein." Advances in Neural Information Processing Systems 32 (2019).

**Questions:**

See above.

---

> ### Author Response · Authors · 2024-11-22
> **Answer to reviewer hY8K**
>
> Dear reviewer, thanks for your feedback. Below we address your questions. A note about the modifications we made in the revised version of our manuscript as well as supplementary materials (code) is available in our general "Modifications" [comment](https://openreview.net/forum?id=sRaAt9OOnW&noteId=PZrOwCkVpG).
>
> ## [Weakness 1:] Soundness of the theoretical claims: A major concern is on the main theorems (5.1, 5.2). These results use standard duality and reformulations that involve the optimal Monge map, but the existence is never justified. In particular, it is well known that GW or inner product GW does not necessarily have optimal map, i.e. the minimum may not be achieved [1]. Same for the existence of the optimal potential, see [2]. Thus the claims mostly only serve as heuristics, as theorem 5.2 is solely on characterization of the optimal map. More need to be discussion regarding why parametrizing by neural network is valid when the optimum may not be achieved.
>
> At first, we agree with the reviewer that our theoretical results are more-or-less standard in the OT community (they follows from the standard duality). Secondly, we thank the reviewer for rising the point on the existence of the GW map. We noticed that in Subsection 5.1 we were a little incorrect when assuming the existence of the optimal map $T^*$. To be more theoretically rigorous, in the revised version of our manuscript (Subsection 5.1) we placed more strict assumptions: (a) compactness of the supports and absolute continuity for the reference measures $\mathbb{P}$, $\mathbb{Q}$; (b) $d_x \geq d_y$ (dimensionality of the source space is not less that the target one). Under these assumptions, the existence of $T^*$ holds (gathered from the papers provided by the reviewer in the second point of the review - we thanks a lot for providing good references).
>
> Note that, in practice, the restriction (a) can always be considered as fulfilled (as we just have some samples from the unknown distributions, which could be confined by some compact set and assumed to be derived from continuous distribution). The restriction (b) is more interesting. Practically speaking, our developed algorithm for NeuralGW can, in principle, work for $d_x < d_y$, but without any guarantee. Furthermore, our practical validations show (see Figure 8 in the Appendix) that when it is the case, i.e., $d_x < d_y$, NeuralGW performs bad. We think this may be due to the gaps in the theory for such cases.
>
>
> ## [Weakness 2:] "Discussion of existing methods and novelty of the proposed algorithm: [...]". Additional references, baselines etc.
>
> We appreciate the question as we find it insightful to strengthen our work. At first, in the revised version of our manuscript we properly cite all the provided relevant works. We thank the reviewer for pointing out some theoretical works, especially (Dumont et. al., 2024). The latter helped us a lot. Also, we found that works (Hur et. al., 2024) and (Zhang et. al., 2022) (the methods are similar) constitute a good continuous GW baseline which we previously missed. We filled this gap in the revised version of our manuscript, entitled the new baseline CycleGW (since it resembles CycleGAN). Please see the updated Section 3 in the revised manuscript. We updated our experimental Section 5.2 correspondingly. Unfortunately, we haven't found CycleGW to work well in practice probably due to the reliance on MMD/Sinkhorn divergence used to ensure the marginal constraints $\mathbb{P}$ and $\mathbb{Q}$ of the learned source$\leftrightarrow$target maps.

---

> ### Author Response · Authors · 2024-11-22
> **Answer to reviewer hY8K. Part 2**
>
> ## [Weakness 3:] Identifying the main issue of discrete GW solvers
>
> We assume (please see the conclusion paragraph of our section 4.2 in our paper) that the main reason of the poor performance of discrete solvers under uncorrelated ($\alpha = 0$) setup is as follows. Small amount of data used for discrete solvers hardly could "catch" the intrinsic geometry of the underlining distribution. In fact, when we apply discrete GW solver, the GW is learned between the geometries induced by sample distributions, not original distributions. These "induced" geometries may be different from the original ones, they may have other symmetries and other properties. Matching them may result in GW map which is quite different from the real map.
>
> On the other hand, when correlatedeness is good ($\alpha = 1$), the GW problem is reduced to finding the proper permutation of the data. The true solution of discrete GW in this case *coincides* with the true underlining GW map. The only remaining question in this case - whether the learned discrete GW map generalizes fairly well for new samples. Our experiments in Section 4.2. demonstrates that it seems to be the case. We prepared a good illustration for the explanation above, please take a look at the anonymized google drive: [link](https://drive.google.com/file/d/11278NcKDhv8SG3nbpNpYFQDQ-SdZQ_NF/view?usp=drive_link).
>
> Also, we added the discussion above to the revised version of our manuscript, please take a look at **Appendix D.1**.
>
> ### Concluding remarks.
>
> Please reply to our post and inform us if the clarifications provided adequately address your concerns regarding our work. We are more than willing to discuss any remaining points during the discussion phase. If the responses offered meet your satisfaction, we kindly request you to consider raising your score.
>
> **References**
>
> Dumont, Théo, Théo Lacombe, and François-Xavier Vialard. "On the existence of Monge maps for the Gromov–Wasserstein problem." Foundations of Computational Mathematics (2024): 1-48.
>
> Zhang, Zhengxin, et al. "Gromov–Wasserstein distances: Entropic regularization, duality and sample complexity." The Annals of Statistics 52.4 (2024): 1616-1645.
>
> Mémoli, Facundo, and Tom Needham. "Comparison results for Gromov–Wasserstein and Gromov–Monge distances." ESAIM: Control, Optimisation and Calculus of Variations 30 (2024): 78.
>
> Mémoli, Facundo, and Tom Needham. "Distance distributions and inverse problems for metric measure spaces." Studies in Applied Mathematics 149.4 (2022): 943-1001.
>
> Hur, YoonHaeng, Wenxuan Guo, and Tengyuan Liang. "Reversible Gromov–Monge sampler for simulation-based inference." SIAM Journal on Mathematics of Data Science 6.2 (2024): 283-310.
>
> Zhang, Zhengxin, et al. "Cycle consistent probability divergences across different spaces." International Conference on Artificial Intelligence and Statistics. PMLR, 2022.
>
> Titouan, Vayer, et al. "Sliced gromov-wasserstein." Advances in Neural Information Processing Systems 32 (2019).
>
> Alexander Korotin, Vage Egiazarian, Arip Asadulaev, \& Evgeny Burnaev (2019). Wasserstein-2 Generative Networks. CoRR, abs/1909.13082.

---

> > ### Comment · Reviewer_hY8K · 2024-11-24
> >
> > I thank the authors for the detailed answer, and I am pleased with the newly added experiments, especially those regarding MUSE dataset. However the discussion over the discrete to continuous issue in the new D.1 seems to be heuristic and does not draw strong connections to learning concepts such as generalization and complexity, thus not too deep theoretically, though I acknowledge the efforts in the newly added discussions. I thus decide to increase the score to 6.

---

> > > ### Author Response · Authors · 2024-12-02
> > > **Thank you!**
> > >
> > > We thank the reviewer for the score increase and for the positive evaluation of our work! While we agree with the reviewer that a thorough analysis of the discrete vs. continuous GW properties from the perspective of statistical learning theory (if it is possible) would appreciate, it would considerably complicate our work which is already full of various experimental setups, comparisons, methodological choices and proposals, etc.

---

### Official Review · Reviewer_9WLQ · 2024-11-04

**Soundness:** 3
**Presentation:** 3
**Contribution:** 2
**Rating:** 5
**Confidence:** 4

**Summary:**

This paper studies the problem of solving the Gromov-Wasserstein (GW) problem numerically. It is shown, in particular, that the case where two datasets are sampled i.i.d. from distributions are particularly challenging for currently available methods.The paper also proposes a neural network-based solver which, in experiments, is seen to outperform other standard methods in the case that correlation between paired training samples is low.

**Strengths:**

The paper accurately identifies a deficiency in certain GW solvers related to training based on paired samples. The proposed neural network-based GW solver appears to be a reasonable way of dealing with this issue as evidenced by the numerical experiments.

**Weaknesses:**

I believe the findings of this paper are not very surprising. Indeed, many of the methods in the paper explicitly assume the setting of paired samples. Furthermore, the GW problem is, in general a computationally hard problem due to its nonconvexity.

Other than the above empirical observation, the paper does not present many results. Indeed, Lemma 5.1 and Theorem 5.2 are well-known in the literature; these are the only theoretical results provided. As such, the proposed neural approach is not coupled with any guarantees, I believe this severely limits the impact of this paper.

Finally, the numerical experiments are interesting, but are not very conclusive. Indeed, only one experiment is provided. On a related note, half of the methods to which the approach is being compared are labelled as "other methods" in the graphs; this is quite baffling to me and should be rectified.

**Questions:**

1. The authors state that some of the methods are trained on 3K samples whereas their method is trained on 200K samples. While it is acknowledged that this yields an unfair comparison, the difference is a consequence of the poor scaling of the other solvers. This begs the question of why experiments on a smaller scale were not performed to compare the methods fairly.

2. It appears that the neural entropic Gromov-Wasserstein solver from T. Wang and Z. Goldfeld, Neural Entropic Gromov-Wasserstein Alignment, arXiv, 2023. would serve as a good method to compare to as no structure is assumed on the data generating process.

---

> ### Author Response · Authors · 2024-11-22
> **Answer to reviewer 9WLQ**
>
> Dear reviewer, thank you for spending time reviewing our paper. We provide answer to your questions and comments below. A note about the modifications we made in the revised version of our manuscript as well as supplementary materials (code) is available in our general "Modifications" [comment](https://openreview.net/forum?id=sRaAt9OOnW&noteId=PZrOwCkVpG).
>
> ## [Weakness 1]: I believe the findings of this paper are not very surprising. Indeed, many of the methods in the paper explicitly assume the setting of paired samples. Furthermore, the GW problem is, in general a computationally hard problem due to its nonconvexity.
>
> On the one hand, we agree with the reviewer that (due to, e.g., computational hardness) one may expect that something may go wrong with existing GW solvers.  However, we humbly think that the findings of our paper are quite nontrivial, sound and deserve the separate research, because:
>
> - We manage to detect the key characteristic of the data (*correlatedeness*) which may drastically affect the performance of several GW solvers; Low correlatedeness may lead to performance deterioration.
>
> - We conduct accurate and extensive benchmark study of existing solvers with *quantitative* estimation of the performance deterioration, i.e., we showcase explicit figures demonstrating *to which extent* the problem is significant.
>
> - We propose a new solver, NeuralGW (which requires theoretical derivation of the objective). It is not ideal, but it demonstrates that the problems with unpairedness could be, in principle, solved.
>
> We believe that it is only through our study that many researchers will notice that many existing GW studies actually use paired data. This information, we agree with the reviewer, is not hidden in these papers. *But it seems that the top importance of this information was not recognized even by the authors of these papers*.
>
> Additionally, if the paired data is accessible, then the problem can simply be reduced to a matching problem, similar to the Wasserstein-Procrustes problem (Grave et. al., 2018), (Aboagye et. al., 2022) or even to more general-purpose data-to-data generative models which have nothing to do with Gromov-Wasserstein and Optimal Transport problems.
>
> ## [Weakness 2]: Other than the above empirical observation, the paper does not present many results. Indeed, Lemma 5.1 and Theorem 5.2 are well-known in the literature; these are the only theoretical results provided. As such, the proposed neural approach is not coupled with any guarantees, I believe this severely limits the impact of this paper.
>
> We agree with the reviewer that our theoretical results are more-or-less standard in the OT community (they follows from the standard duality). Also, we agree that our adversarial objective (6) is indeed not guaranteed to recover an optimal GW map (the only we know is that an optimal map is a solution of our adversarial objective - see Theorem 5.2, not vice versa). We hypothesize that it may be complicated to develop further theoretical guarantees of NeuralGW due to following reasons:
>
> - Possible non-uniqueness of Gromov-Wasserstein optimal maps (if the source and target distributions possess some symmetries, these symmetries preserve the optimality of GW map).
>
> - Possible presence of *fake solutions* of semi-dual OT formulation (eq. (8) in Appendix A), see Section 3.1 of (Korotin et. al., 2023a). Note that eq. (8) plays key role in deriving our objective eq. (6).  The problem is that map $T^*$ which solves eq. (8) is not guaranteed to recover OT map between $\mathbb{P}$ and $\mathbb{Q}$. Importantly, the presence of *fake solutions* does not stop researches from proposing efficient neural OT solvers based on semi-dual OT, e.g., (Korotin et. al., 2023b), (Fan et. al., 2023).
>
> Anyway, the primary goal of our paper is to propose a benchmark that explores some of the problems and gaps present in the evaluation of the GW solvers (mainly related to uncorrelatedness). The NeuralGW solver is included as an attempt to alleviate the dependence of the solvers to correlated (or paired) data. It is not perfect in theoretical sense, but it is the only method capable of approximating the geometry of distributions even in completely uncorrelated setups.

---

> ### Author Response · Authors · 2024-11-22
> **Answer to reviewer 9WLQ. Part 2**
>
> ## [Weakness 3]: Finally, the numerical experiments are interesting, but are not very conclusive. Indeed, only one experiment is provided. On a related note, half of the methods to which the approach is being compared are labelled as "other methods" in the graphs; this is quite baffling to me and should be rectified.
>
> We refer to the second point of our [general response](https://openreview.net/forum?id=sRaAt9OOnW&noteId=wdEmixZ12I) to check additional experiments where we include a new dataset and a new method of embedding.
>
> Regarding "other methods" label in Figure 5 - we think there is a tiny misunderstanding. Note that the primary aim of Figure 5 is to compare methods trained on large amount of data $N_{train} = 360K$ (note that in the original manuscript we mistakenly wrote $N_{train} = 400K$, sorry for this - we fixed this mistake in the revised manuscript), see lines 462-464 in our text. The only methods which are capable dealing with such amount of data are our NeuralGW, RegGW, newly added (during the rebuttal) CycleGW and also the adapted minibatch version of FlowGW. The methods in gray (labelled as "other methods") **are trained on 6K samples**, the results for these small-dataset approaches are provided only for reference. See lines 470-473 for the explanation.  We consider that adding labeled information for these methods might be confusing for most of the readers.
>
> Note that the colored and labeled versions of the grayed plotted lines in Figure 5 can be found in Appendix B.1, Figure 7.
>
> ##  [Question 1]: The authors state that some of the methods are trained on 3K samples whereas their method is trained on 200K samples. While it is acknowledged that this yields an unfair comparison, the difference is a consequence of the poor scaling of the other solvers. This begs the question of why experiments on a smaller scale were not performed to compare the methods fairly.
>
> We refer to Table 2 in the Appendix, which presents the results for NeuralGW trained on a small dataset (3K samples). The table clearly demonstrates that the method fails to perform effectively with such limited data. This limitation, however, is inherently tied to the nature of the solver, as it is composed of three neural networks, which are typically data-intensive.
>
> ## [Question 2]: It appears that the neural entropic Gromov-Wasserstein solver from T. Wang and Z. Goldfeld, Neural Entropic Gromov-Wasserstein Alignment, arXiv, 2023. would serve as a good method to compare to as no structure is assumed on the data generating process.
>
> Thank for suggesting an additional approach to strengthen our paper, we cited it in the revised version of our manuscript. In fact, we have already considered the addition of (Wang et. al, 2023), however, it proved to be difficult due to the absence of an official implementation.
> At the same time, we read the proposed paper and found several drawbacks for the solver, we list some of them below:
>
> - The algorithm proposed in the (Wang et. al, 2023), (Algorithm 1) seems to be suitable only for large values of entropic regularization parameter epsilon (as the authors point out in Section 4.1 of their paper). While many of our considered GW methods are also based on entropic GW, too large entropic regularization may introduce noticeable biases which spoil the performance.
>
> - The neural estimator introduced in (Wang et. al, 2023, Section 4.1) is a biased estimator since it takes the logarithm of an averaged sum of exponential terms for every batch instead of the whole dataset. This sum may behave unstably.
>
> - Although the method uses neural networks, it does not provide a way to do out-of-sample estimation for GW map (or plan). As we understand, the method results in the coupling matrix (Wang et. al., 2023 , eq. 22) which does not generalize to new (unseen) samples.
>
> Based on this, we humbly think that the solver from (Wang et. al., 2023) could be excluded from our comparison.
>
>
> ### Concluding remarks.
>
> Please reply to our post and inform us if the clarifications provided adequately address your concerns regarding our work. We are more than willing to discuss any remaining points during the discussion phase. If the responses offered meet your satisfaction, we kindly request you to consider raising your score.

---

> > ### Author Response · Authors · 2024-11-22
> > **References**
> >
> > Grave E, Joulin A, Berthet Q. 2018. Unsupervised alignment of embeddings with Wasserstein Procrustes. arXiv:1805.11222 [cs.LG]
> >
> > Aboagye, P., Zheng, Y., Yeh, M., Wang, J., Zhuang, Z., Chen, H., Wang, L., Zhang, W., Phillips, J. (2022). Quantized Wasserstein Procrustes Alignment of Word Embedding Spaces. In Proceedings of the 15th biennial conference of the Association for Machine Translation in the Americas
> >
> > Korotin, Alexander, Daniil Selikhanovych, and Evgeny Burnaev. "Kernel Neural Optimal Transport." The Eleventh International Conference on Learning Representations. 2023a.
> >
> > Korotin, Alexander, Daniil Selikhanovych, and Evgeny Burnaev. "Neural Optimal Transport." The Eleventh International Conference on Learning Representations. 2023b.
> >
> > Fan, Jiaojiao, et al. "Neural Monge Map estimation and its applications." Transactions on Machine Learning Research. 2023.
> >
> > Wang, Tao, and Ziv Goldfeld. "Neural Entropic Gromov-Wasserstein Alignment." arXiv preprint arXiv:2312.07397 (2023).

---

> > > ### Comment · Reviewer_9WLQ · 2024-11-24
> > >
> > > I thank the authors for their detailed response. I will increase my score, as additional experiments have been added and my questions were adequately answered. However, I still believe that the findings of this paper are not very surprising.

---

> > > > ### Author Response · Authors · 2024-11-25
> > > > **Thank you**
> > > >
> > > > We are pleased that our responses and efforts have been recognized. However, while we respect the reviewer's opinion, the soundness of our results is the subject of our disagreement with the reviewer.

---

### Official Review · Reviewer_wk3d · 2024-11-04

**Soundness:** 2
**Presentation:** 3
**Contribution:** 2
**Rating:** 6
**Confidence:** 3

**Summary:**

The paper discusses the challenges of solving the Continuous Gromov-Wasserstein Optimal Transport (GWOT) problem and emphasizes the need for a reliable and general-purpose method to address these issues. The authors claim that current solvers rely on discrete methods and struggle when dealing with uncorrelated data. They then introduce NeuralGW, which is claimed as a method that does not depend on discrete approximations and can therefore handle realistic scenarios with uncorrelated data. However, their method is unstable and requires a large amount of data for training.

**Strengths:**

- Analysis of existing GWOT solvers.
- Fill a gap in GWOT literature by questioning the assumptions under which existing solvers operate.
- Proposed partial solution for defined issues.

**Weaknesses:**

- `Diversity`: The evaluation focuses on the GloVe dataset and the biological dataset. It's better to expand experiments to cover additional domains.
- ~The paper still has some issues that need to be addressed in order to support its claims~
- Lack of experiment setup details.

**Questions:**

- Given the high data requirements of NeuralGW's adversarial optimization, would you consider alternate training methods or adaptations to improve performance on smaller datasets? Further clarification on the trade-offs between data size and performance stability would also be insightful.
- Would it be possible to see an empirical or theoretical comparison between NeuralGW and other continuous GWOT methods, particularly on tasks where preserving intra-domain structure is less critical?
- Line 311, indices from the target are incorrect.
- Why were Top k-accuracy, cosine similarity, and FOSCTTM chosen for evaluation?

---

> ### Author Response · Authors · 2024-11-22
> **Answer to reviewer wk3d**
>
> Dear reviewer. Thank you for your review and for noticing our efforts and importance of our contributions. Please find below our comments addressing your concerns. A note about the modifications we made in the revised version of our manuscript as well as supplementary materials (code) is available in our general "Modifications" [comment](https://openreview.net/forum?id=sRaAt9OOnW&noteId=PZrOwCkVpG).
>
> ## [Weakness 1]: Diversity of experiments.
>
> We understand the importance of showing additional comparisons in different scenarios, therefore, we provide additional results for three new setups, please find them in the second point of our [general response](https://openreview.net/forum?id=sRaAt9OOnW&noteId=wdEmixZ12I).
>
> Even though these additional results explore new domains which are not too far from what we had previously
> (since we stick to word and sub-word embeddings), the new experiments highlight the impact of embedding types on the construction of datasets to be aligned. This insight is crucial for understanding the potential of GW solvers, their limitations, and possible strategies for constructing datasets that can be effectively aligned with GW-like approaches.
>
> Additionally, as our work aims to provide a clear and fair comparison between GW solvers of different natures (discrete and continuous), we are constrained to work with datasets in which we know the ground truth pairs and also have a suitable number of samples to train the batched methods (>40K, depending on the embedding method). Dealing with datasets where the true pairs are not known is complicated due to the absence of proper quality assessment metric. All this makes it difficult to increase the diversity of the benchmark setups considered.
>
> ## [Weakness 2]: Issues to support the claims.
>
> Thanks for the feedback. We kindly ask you to specify the issues that need to be clarified so we can provide the necessary additions/clarifications or revisions.
>
> ## [Weakness 3]: Lack of experimental setup details.
>
> We thank the reviewer for the feedback. However, we humbly think that it is not quite accurate statement that our used experimental setups lack the details. In fact, each of our experimental subsection (4.2) and (5.2) is started with carefully introduced explanation of the experimental setup (also, we expanded Section 4.2 and the Appendix with newly added BPEmb experiments). We have section B in the Appendix with carefully described metrics and expanded with BPEmb experiment peculiarities. Also, for the ease of navigation, **we add Table 1 summarizing our considered setups**. We hope that our answer clarifies the situation with the setups' details. We are ready to answer the remaining questions here.

---

> ### Author Response · Authors · 2024-11-22
> **Answer to reviewer wk3d (Part 2)**
>
> ## [Question 1]: Given the high data requirements of NeuralGW's adversarial optimization, would you consider alternate training methods or adaptations to improve performance on smaller datasets? Further clarification on the trade-offs between data size and performance stability would also be insightful.
>
> While alternative methods can be used to train NeuralGW on smaller datasets, the amount of data is crucial not only for the solver to work effectively but also to accurately capture the underlying geometry of the distributions. This importance is demonstrated in Appendix B.1 (Experiment 2), where baseline solvers were trained with 3,000 samples. Although their performance was good when evaluated on a restricted space (~8,000 samples), it dropped significantly when the evaluation was extended to the full reference space of 400K samples.
>
> Additionally, the dependence between dataset size and performance for NeuralGW can also be mitigated by selecting a more suitable embedding method, as shown in our additional experiments (see the second point of our [general response](https://openreview.net/forum?id=sRaAt9OOnW&noteId=wdEmixZ12I)). In particular, the third additional experiment involved a reduced Twitter dataset embedded using Byte-Pair embeddings (BPEmb). NeuralGW demonstrated a robust performance in this setup despite the significant reduction in terms of training samples (from 180K in the case of GloVe to 45K for BPEmb).
>
> We also conducted an additional experiment in which we use 45K Twitter-GloVe samples to have a fair comparison with respect to the Twitter-BPEmb dataset used in the additional experiments, find the results below (the dimensionalities are $100 \rightarrow 50$):
>
> | Corpus - Type of emb.                 | Training Size | $\alpha$ | Top1   | Top5   | Top10   | Cosine Similarity | FOSCTTM  |
> |-------------------------|---------------|----------|--------|--------|---------|-------------------|----------|
> | Twitter - GloVe         | 45K           | 1.0      | 0.001  | 0.003  | 0.005   | 0.2578            | 0.3854   |
> | Twitter - BPEmb         | 45K           | 1.0      | 0.7261 | 0.7356 | 0.74001 | 0.7726            | 0.1463   |
>
> As we can see, the performance of NeuralGW trained on 45K GloVe embeddings vs. 45K BPEmb embeddings considerably decreases.
>
> Therefore, as a general answer to the question, we can conclude that it is more suitable for our NeuralGW solver to find a more appropriate embedding method rather than providing more data. This not only improves the performance in terms of the metrics but it also alleviates the dataset size dependence.
>
> ## [Question 2]: Would it be possible to see an empirical or theoretical comparison between NeuralGW and other continuous GWOT methods, particularly on tasks where preserving intra-domain structure is less critical?
>
> We humbly believe that providing such experiments is out of the scope of our paper. At first, it would require to redefine the ultimate construction of our benchmark, e.g., some of our key concepts like correlatedness will become unnecessary in some parts of our text. This will lead to broken logic of our message and fragmented exposition of our results. Secondly, the requested additional experiments would require to redefine our metrics as the task will be slightly modified (some metrics might not be suitable). This will further complicate the content of the paper. Finally, if keeping the intra-domain distances is not critical, other type of solvers might be more suitable, i.e., standard data-to-data solvers based on adversarial training., e.g., StarGAN (Choi et. al., 2018), or diffusion models, e.g., DDBM (Zhoi et. al., 2024).
>
>
> ## [Question 3]: Line 311, indices from the target are incorrect.
>
> Thanks for the observation , the correct range should be from $(1-\alpha)N_{train}/2$ to $(1-\alpha/2)N_{train}$, we made this fix in the revised manuscript.

---

> ### Author Response · Authors · 2024-11-22
> **Answer to reviewer wk3d (Part 3)**
>
> ## [Question 4]: Why were Top k-accuracy, cosine similarity, and FOSCTTM chosen for evaluation?
>
> We considered these metrics due to their use in other papers dealing with GW solvers: Top-k accuracy was used in (Alvarez-Melis et.al., 2018), FOSCTTM in (Klein et. al., 2024) and (Demetci et. al., 2020). Cosine similarity is just quite natural to operate with word embeddings (Mikolov et. al., 2013). Importantly, all of these metrics are familiar/intuitive for the community.
>
> While other metrics are also possible, we humbly think that the available ones already provide a full scope of the capabilities of the solvers needed for the comparison.
>
> ### Concluding remarks.
>
> Please reply to our post and inform us if the clarifications provided adequately address your concerns regarding our work. We are more than willing to discuss any remaining points during the discussion phase. If the responses offered meet your satisfaction, we kindly request you to consider raising your score.
>
> **References**
>
>
> David Alvarez-Melis and Tommi Jaakkola. Gromov-wasserstein alignment of word embedding spaces. In Proceedings of the 2018 Conference on Empirical Methods in Natural Language Processing. 2018.
>
> Dominik Klein, Théo Uscidda, Fabian Theis, and Marco Cuturi. Generative entropic neural optimal transport to map within and across space, 2024
>
> Demetci, P. Santorella, R. Sandstede, B., Noble, W. S., Singh, R. Gromov-Wasserstein based optimal transport for aligning single-cell multi-omics data. 2020.
>
> Choi, Yunjey, et al. "Stargan: Unified generative adversarial networks for multi-domain image-to-image translation." Proceedings of the IEEE conference on computer vision and pattern recognition. 2018.
>
> Zhou, Linqi, et al. "Denoising Diffusion Bridge Models." The Twelfth International Conference on Learning Representations. 2024.
>
> Mikolov, Tomas. "Efficient estimation of word representations in vector space." arXiv preprint arXiv:1301.3781 3781 (2013).

---

> ### Comment · Reviewer_wk3d · 2024-11-25
>
> I appreciate the authors for their detailed response, and I am pleased with the new experiments, particularly those involving the MUSE dataset. Therefore, I have decided to increase the score to 6.

---

> > ### Author Response · Authors · 2024-11-25
> > **Thank you**
> >
> > We thank the reviewer for the positive feedback! We are glad that our answers and efforts were appreciated.

---

### Author Response · Authors · 2024-11-22
**Modifications**

**Additions/changes in the revised version of our manuscript**

- **(all reviewers)** We provide a revised version of the paper with additions and modifications, we marked all these changes with blue colored text.

- **(all reviewers)** In the original manuscript we used GloVe to name the dataset as well as the type of embeddings, in the revised paper we expand the type of embeddings used, therefore, this old notation may be confusing. Now we are using Twitter-GloVe to refer to the Twitter corpus embedded using GloVe algorithm, so does for other datasets and embedding algorithms. Please see Section 4.2 and Appendixes B.1, B.2 of the revised version.

- **(all reviewers)** The captions of tables and figures were modified to make them self-explainable, we decided this due to the number of experiments we currently have in the paper.

- **(all reviewers)** Citations of the mentioned papers during the rebuttal were added.

- **(hY8K)** In Section 3 we added a short paragraph explaining a new type of solver we tested on our setup, we refer to this solver as CycleGW. Please, find more details in the aforementioned section.

- **(hY8K)** Section 5.1 (theoretical derivation of NeuralGW) is a bit modified.

- **(wk3d, 9WLQ)** Section 4.2 is expanded with new BPEmb embedding explanation. New Appendix B.2 is added with the details and experiments with BPEmb.

- **(hY8K, ctGz, 4w5B)** New Appendix D.1, explaining the issue of discrete GW solvers.

**Code availability.**


- Our provided code contains implementations of **all the baselines considered in the paper** to ease the reproducibility of our results. In the revised version, we updated the codebase with newly considered baseline (CycleGW).

- Also, our revised version of the code contains all necessary infrastructure for running newly added BPEmb experiments.

---

### Author Response · Authors · 2024-11-22
**General Response**

We thank the reviewers for their time and feedback. We appreciate that the reviewers recognize the importance of the problem under consideration, the significance of the limitation of GW solvers revealed by our paper, and the novelty of our NeuralGW solver. Here we make some general comments as well as share some additional results.

## 1. General clarification about continuous solvers results and experiments (all reviewers)

In the original version of our manuscript we conducted some experiments in a somewhat wrong manner, more specifically we refer to  the experiments in Figure 5 (main text, original manuscript) and Figure 8 (in the Appendix, original manuscript) for the continuous/batched solvers (NeuralGW, FlowGW\_mb, and RegGW). Our mistake was on the use of hard-coded source-target relationships (i.e., there were one-to-one correspondence between source embeddings and target embeddings picked to form a batch for training), what leaded to emulate a paired setup.
This explains the significant performance increase observed when the correlatedeness changes from any $\alpha<1$ to $\alpha = 1$. The original plots highlighted both the effectiveness of the methods and the correctness of our implementations under this ill-posed scenario. However, it is essential to provide a revised version of these experiments that follows the pipeline proposed in Section 4.1, which was used for benchmarking the baseline solvers. In this setup, the source and target batches for training are selected independently. The updated Figures 5 and 8 are available in the revised version of our manuscript and substitute the original ones. The updated plots demonstrate that under proper setup, the performance "explosion" is no longer observed.


It is also necessary to clarify that the exact number of samples we used for training the continuous methods is 360K instead of the 400K we initially stated in the paper, we updated this information in the revised version.

## 2. Additional experimental results (reviewers wk3d and 9WLQ)

Here we provide additional results for the GW solvers tested on the paper. Two reviewers (wk3d and 9WLQ) expressed their concerns about this fact, however, we find these new results to be relevant and insightful for all the reviewers.

In the new experiments we use a new type of embeddings called Byte-Pair embeddings (Heinzerling et. al., 2018)  (BPEmb), which are sub-word embeddings. The detailed information is given in updated Section 4.2 and newly added **Appendix B.2** of our revised manuscript. The authors provide pre-trained versions that we used to embed different datasets, in all the cases the vocabulary size considered was 200K. The words from the datasets that are not explicitly contained in the original vocabulary were excluded from the experiments.

We provide **three** additional experiments listed below:

- MUSE dataset (Conneau et. al., 2017) with BPEmb (90K samples). Languages: English $\rightarrow$ English. Dimensions: 100  $\rightarrow$ 50. See Figure 9, Appendix B.2.

- MUSE dataset with BPEmb (60K samples). Languages: English  $\rightarrow$  Spanish. Dimensions: 100 $\rightarrow$ 100. See Table 3, Appendix B.2.

- Twitter dataset with BPEmb (90K samples). Dimensions: 100  $\rightarrow$  50. See Figure 10, Appendix B.2.


To help the reviewers and readers with navigating over all our various conducted experiments (original and newly conducted), in Appendix B we added Table 1 summarizing all the considered experimental setups.

**Conclusion:** These new experiments demonstrate that the performance of the solvers —both baselines and continuous solvers— improved, with increased accuracy and reduced dependence on correlated data. However, for lower correlations ($\alpha<0.4$), the results remain consistent with previous experiments, where NeuralGW continues to be the only viable solver for uncorrelated and partially correlated scenarios.

**References.**

Benjamin Heinzerling, \& Michael Strube (2018). BPEmb: Tokenization-free Pre-trained Subword Embeddings in 275 Languages. In Proceedings of the Eleventh International Conference on Language Resources and Evaluation (LREC 2018).

A. Conneau*, G. Lample*, L. Denoyer, MA. Ranzato, H. Jégou, Word Translation Without Parallel Data, 2017.

---

### Meta-Review · Area_Chair_s8Ui · 2024-12-17

**Metareview:**

The paper addresses the continuous Gromov-Wasserstein Optimal Transport (GWOT) problem and proposes a benchmark alongside a NeuralGW solver to mitigate issues in existing methods. Despite the novelty of focusing on continuous GWOT, the reviewers highlighted several weaknesses that persisted even after the rebuttal. A major concern is the lack of clear motivation and relevance for machine learning (ML) applications. The paper does not convincingly show how the findings provide new insights into ML problems or improve practical scenarios where OT is applied. The theoretical claims, such as the existence of the Gromov-Monge map, are based on heuristics and remain unsubstantiated for non-compact or asymmetric domains. Experimentally, the benchmarks are narrow, focusing on limited datasets (e.g., GloVe embeddings), and fail to include real-world tasks relevant to ML. Additionally, the NeuralGW solver shows poor scalability and high sensitivity to hyperparameters, requiring large datasets to achieve meaningful results. While some new experiments were added, they primarily reinforce the paper’s findings without addressing these broader concerns. The combined lack of theoretical depth, practical relevance, and diversity in experimentation limits the paper’s contributions.

**Additional Comments On Reviewer Discussion:**

During the rebuttal, reviewers raised issues regarding the limited experimental setups, unverified theoretical assumptions, and weak connections to ML relevance. While the authors added experiments and improved theoretical explanations, they failed to address the key concern of practical impact and broader applicability. These unresolved weaknesses were central to the decision to recommend rejection.

---

### Decision · Program_Chairs · 2025-01-22

Reject